# Endothelial *Jak3* expression enhances pro-hematopoietic angiocrine function in mice

José Gabriel Barcia Durán [1], Tyler Lu[1,2], Sean Houghton [1], Fuqiang Geng[1], Ryan Schreiner [1,3], Jenny Xiang[4], Shahin Rafii [1,2], David Redmond [1✉] & Raphaël Lis [1,2✉]

Jak3 is the only non-promiscuous member of the Jak family of secondary messengers. Studies to date have focused on understanding and targeting the cell-autonomous role of Jak3 in immunity, while functional *Jak3* expression outside the hematopoietic system remains largely unreported. We show that *Jak3* is expressed in endothelial cells across hematopoietic and non-hematopoietic organs, with heightened expression in the bone marrow. The bone marrow niche is understood as a network of different cell types that regulate hematopoietic function. We show that the *Jak3⁻/⁻* bone marrow niche is deleterious for the maintenance of long-term repopulating hematopoietic stem cells (LT-HSCs) and that *JAK3*-overexpressing endothelial cells have increased potential to expand LT-HSCs in vitro. This work may serve to identify a novel function for a highly specific tyrosine kinase in the bone marrow vascular niche and to further characterize the LT-HSC function of sinusoidal endothelium.

[1] Ansary Stem Cell Institute, Division of Regenerative Medicine, Department of Medicine, Weill Cornell Medicine, New York, NY 10065, USA. [2] Ronald O. Perelman and Claudia Cohen Center for Reproductive Medicine and Infertility, Weill Cornell Medicine, New York, NY 10065, USA. [3] Department of Ophthalmology, Margaret Dyson Vision Research Institute, Weill Cornell Medicine, New York, NY 10065, USA. [4] Genomics Resources Core Facility, Weill Cornell Medicine, New York, NY 10065, USA. ✉email: dar2042@med.cornell.edu; ral2020@med.cornell.edu

Adult hematopoietic stem cells (HSCs) reside primarily in the bone marrow, where interactions with the surrounding microenvironment modulate their fate decisions[1]. HSC quiescence, for instance, can be disrupted by said microenvironment or niche[2]. Some of the earliest research in this field showed that radiation-induced myeloablation induces the recruitment of Mmp9 to the membrane of bone marrow niche cells[3]. Mmp9 was found to cleave membrane-bound Scf, releasing it in its soluble form. Increased bio-available Scf, in turn, promoted the expansion and differentiation of quiescent HSCs for hematopoietic recovery. Since then, efforts to identify and interrogate the various cellular players and their molecular interactions in an attempt to define the HSC niche have proven a fertile ground for investigation[4].

The osteogenic niche was one of the first to be linked to hematopoiesis[5,6], but in vivo imaging studies have since revealed that HSCs rarely localized to the vicinity of osteoblastic cells[7,8]. In comparison, stromal cell populations marked by Nestin or Lepr expression have been implicated more thoroughly in HSC localization and function. Nestin-expressing cells secrete Scf and Cxcl12, now known as the two major HSC chemoattractant and survival factors[9–11]. Nestin-cre^ERT2 knock-in mice present significant reductions in bone marrow HSCs, as measured in vitro by colony-forming unit (CFU) assays in a limiting-dilution manner[12]. Lepr[+] bone marrow stroma is a major producer of Scf as well, and Lepr-cre; Scf^fl/GFP knockout mice also exhibit HSC depletion[13]. Even the hematopoietic progeny of HSCs themselves constitute another fraction of their microenvironment[2]. Caspase-activated mosaic ablation of megakaryocytes has been associated with HSC proliferation and exhaustion[14–16], and Cd169[+] macrophages have been shown to promote Cxcl12 secretion in Nestin[+] cells, ultimately leading to HSC mobilization into the circulation[17].

Endothelial cells (ECs), in contrast, comprise the only type of bone marrow niche cell that is able to maintain and expand HSCs in vitro, by sustained production of cytokines such as Scf, Flt3, and Csf2[9,18]. This phenotype has been shown to be contact-dependent, since Notch1[−/−]; Notch2[−/−] HSCs fail to expand in co-culture with bone marrow ECs (BMECs)[19,20]. Growing research of the pro-hematopoietic milieu provided by ECs, or vascular niche, has led to the coinage of angiocrine signaling as its operative term[21], and, more tangibly, to the development of a tractable in vitro platform to expand long-term repopulating HSCs (LT-HSCs) under xenobiotic-free conditions[22].

Angiocrine function has been found to vary from organ to organ[23–25]. The expansion and maintenance potentials of ECs derived from the bone marrow, a primary hematopoietic organ, are higher than those of neighboring stromal cells or ECs derived from other vascular beds, such as the lungs[26]. Even co-infusion of BMECs along with HSCs confers a radioprotective effect post-myeloablation[27]. ECs from the lungs[28] and the liver[29,30] also display similar, organotypic regenerative function. Remarkably, ECs derived from non-hematopoietic tissues, like the central nervous system, also display pro-hematopoietic ability in vitro;[31,32] moreover, brain ECs have been shown to expand HSCs in a contact-independent manner by pleiotrophin secretion[33]. Knowledge of the role that angiocrine signaling plays in HSC maintenance has led to intensive characterization of the arteries and sinusoids that make up the bone marrow vascular niche. Fenestrated bone marrow sinusoids are considered to be VE-Cad[+]CD31[+]Sca1[lo]Vegfr3[+] and abundant, while arterial ECs are defined by VE-Cad[+]CD31[+]Sca1[hi]Vegfr3[−] expression and are less numerous or permeable than their venous counterparts[26,34,35]. HSCs are broadly distributed close to sinusoids, and some imaging-based studies have been quick to claim that HSCs reside in a "peri-sinusoidal" niche[36]. However, endothelial Scf and Cxcl12 are both secreted primarily by bone marrow arterioles[37,38], suggesting that quiescent HSCs may localize to this microenvironment instead.

Despite increasing characterization of secreted factors and surface markers produced by ECs, the intracellular landscape of the bone marrow vascular niche remains largely unexplored. Here, we report the expression of Jak3, a secondary messenger, in the endothelial compartment of the bone marrow microenvironment. Jak3 is one of the four Janus kinases, a family of non-receptor tyrosine kinases involved in multiple cytokine-mediated aspects of immunity[39]. First cloned from human natural killer cells[40], JAK3 possesses two distinctive qualities for a Janus family kinase: (i) it selectively associates with the common cytokine receptor sub-unit γc[41,42] and (ii) it is not ubiquitously expressed[40]. JAK3 mutations have been associated with hematological malignancies[43] and Jak3-null mice display a SCID phenotype[44]. As a result, the study of JAK3 has been focused on hematopoietic tissues, although its expression has also been reported in osteoblasts[45], smooth muscle cells[46], and solid tumors[47]. The presence of Jak3 in endothelium has been reported as well[48], but a thorough assessment of EC-specific function has been lacking. We show that Jak3 is expressed across all vascular beds, though more so in venous ECs of sinusoidal morphology that are abundant in primary and secondary hematopoietic organs. Our data show that Jak3 expression is associated with vascular niches that promote LT-HSC maintenance and expansion. As a result, we predict that Jak3 is only one piece of the larger puzzle that attributes specialized ECs their pro-hematopoietic function.

## Results

**Jak3 expression in vitro correlates with pro-hematopoietic culture conditions.** To ascertain the transcriptomic signature of BMECs, we performed comparative bulk RNA-seq analyses. Transcript quantification of primary Ter119[−]CD45[−]CD31[+]VE-Cad[+] cells from the liver, lungs, heart, kidneys, and bone marrow, respectively, showed a reliably vascular transcriptomic signature across all samples (Fig. 1a). In addition to Pecam1 (CD31) and Cdh5 (VE-Cad), our dataset was characterized by consistent expression of other canonical EC surface markers such as Kdr, Cldn5, and S1pr1, as well as angiocrine factors like Egfl7, Igfbp4, Dll4, and Kitl. ETS transcription factors Ets1, Ets2, Erg, and Fli1 were found across all EC types, too. A number of these transcripts are also expressed by hematopoietic cells, but our dataset was comprised of ECs exclusively, since surface markers restricted to the hematopoietic system such as Spn and Csfr3 or transcription factors like Runx1 or Gfi1 were effectively absent from all samples. This expression pattern was congruent with the expected bulk transcriptomic profile of any purified, adult vascular bed without hematopoietic contamination[23].

The transcriptomic distance between ECs from different organs was apparent by principal component analysis (PCA), where PC2 separated BMECs from their liver, lung, heart, and kidney counterparts, accounting for 21.5% of the total variance (Supplementary Fig. 1a). More specifically, differential expression analyses of these data, systematically comparing each vascular bed against the rest, yielded hundreds of unique organotypic gene transcripts per group (Fig. 1b). Plotting the normalized read counts showed that Jak3 was among the gene transcripts uniquely enriched in the bone marrow vascular niche (Supplementary Fig. 1b). Concomitantly, we observed restricted expression of known organotypic EC markers in each of the remaining groups: Gata4 expression was increased in liver ECs[25], Sftpd was enriched in lung ECs[49], Rbp7 was most highly expressed in heart ECs[50,51], and kidney ECs were the only group expressing Slc14a1[52]

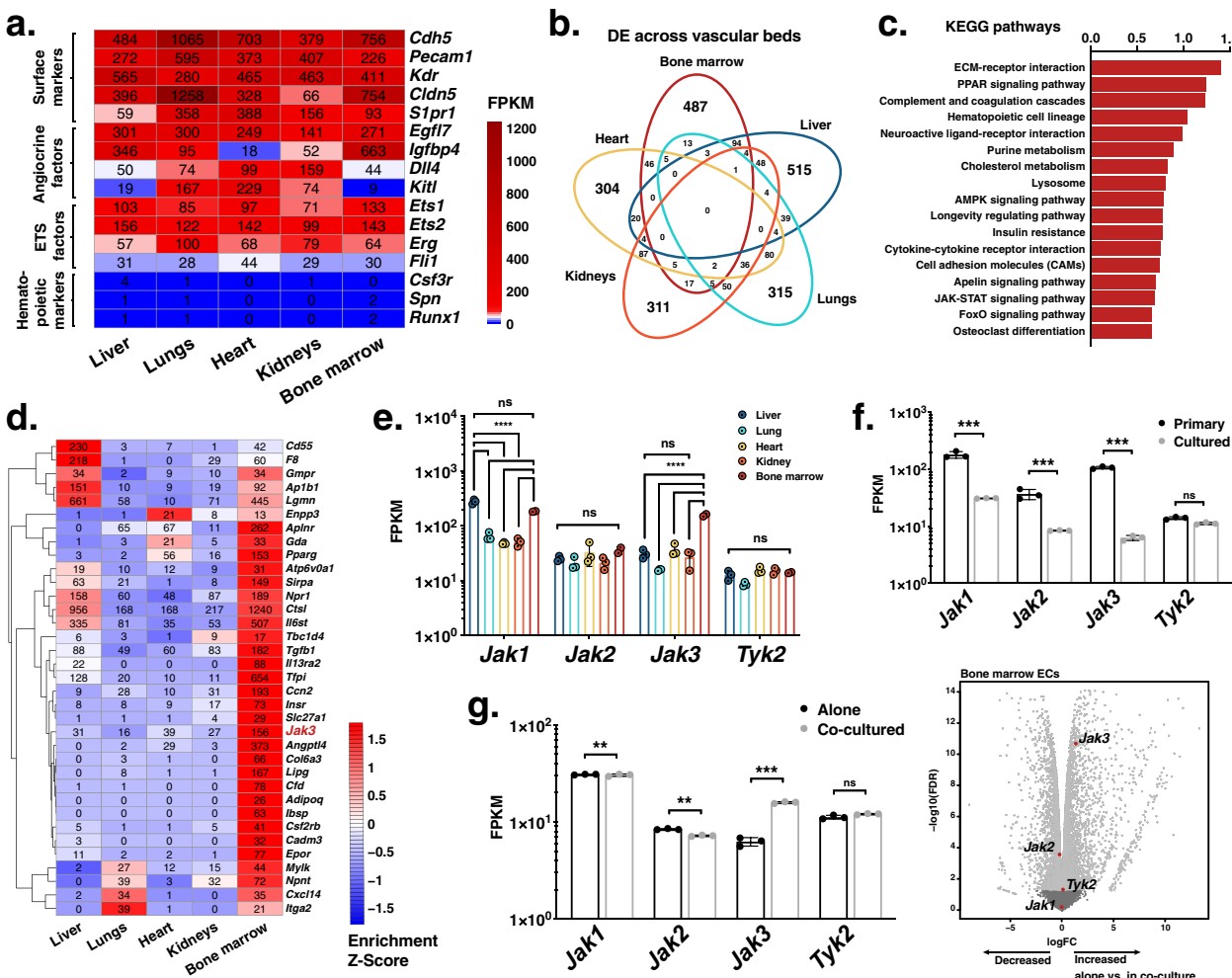

**Fig. 1 Endothelial expression of *Jak3* is enhanced by a hematopoietic milieu in vitro and in vivo. a** Heatmap depicting absolute transcript abundances of genes of interest obtained by RNA sequencing (RNA-seq) of primary murine endothelial cell (EC) populations extracted from the indicated vascular beds. Average FPKM values for all biological replicates ($n = 2$ or 3 per organ) appear in black. **b** Venn diagram showing number of intersecting overrepresented gene transcripts resulting from differential expression analyses of each vascular bed against all other samples (FDR < 0.05). **c** Bar plot of enrichment scores of the 15 most upregulated Kyoto Encyclopedia of Genes and Genomes (KEGG)[54–56] terms generated by gene set enrichment analysis (GSEA) using the $\log_2$ fold change of the differentially expressed gene transcripts in the bone marrow vasculature compared to that of the liver, lungs, heart, and kidneys. **d** Heatmap depicting normalized transcript abundances of genes that contributed to the 15 most upregulated KEGG terms by GSEA. Average FPKM values for all biological replicates ($n = 2$ or 3 per organ) appear in black. *Jak3*, the most highly and differentially expressed transcript, is highlighted in red. **e** Janus kinase (Jak) family gene expression across the organs assayed as measured by RNA-seq. ****$p$ value < 0.0001 and ns (not significant) > 0.05, by two-way ANOVA. **f** Jak family transcript expression in primary vs. cultured bone marrow ECs (BMECs) as measured by RNA-seq ($n = 3$ per group). Raw data were obtained from Poulos et al.[26] (GEO: GSE61636) and re-analyzed for this publication. ***$p$ value < 0.001 and ns > 0.05, by unpaired, two-tailed Student's $t$ test. **g** Jak family transcript expression in BMECs cultured alone vs. in contact with hematopoietic stem and progenitor cells for 8 days as measured by RNA-seq ($n = 3$ per group; left). Volcano plot illustrating differential expression analysis of these data (right). Jak family gene transcripts are highlighted with enlarged dots in red; respectively, light or dark gray dots indicate gene transcripts differentially expressed above or below cutoffs, $-\log_{10}$ ($p$ value) < 1 and $|\log_2 FC|$ < 2.5. Raw data were obtained from Poulos et al.[26] (GEO: GSE61636) and re-analyzed for this publication. ***$p$ value < 0.001, **$p$ value < 0.01, and ns (not significant) > 0.05, by unpaired, two-tailed Student's $t$ test. On all plots, each replicate is represented by an individual dot and error bars represent standard deviation from the mean.

(Supplementary Fig. 1b). As a result, *Jak3* emerged as a potential marker of bone marrow endothelium.

To further interrogate the transcriptional identity of the bone marrow vascular niche in light of this observation, we ran unbiased analyses using a number of functional databases. First, differentially expressed transcripts in BMECs were subjected to gene set enrichment analysis (GSEA)[53] against the Kyoto Encyclopedia of Genes and Genomes (KEGG) database[54–56], revealing enrichment of PPAR, AMPK, and JAK-STAT signaling pathways, among others, among the top 15 terms retrieved (Fig. 1c). Of all the genes that contributed these KEGG terms,

*Jak3* was one of the most highly and uniquely expressed by BMECs with respect to other vascular beds (Fig. 1d). The JAK-STAT signaling pathway was retrieved by GSEA due to overrepresentation of five transcripts in BMECs: *Il13ra2*, *Csf2rb*, *Epor*, *Jak3*, and *Il6st*. A connectivity plot rendered by running these five genes against a database of known and predicted protein–protein interactions called STRING (Search Tool for the Retrieval of Interacting Genes/Proteins)[57] showed the *Jak3* node at the center (Supplementary Fig. 1c). While *Jak1*, *Jak2*, and *Tyk2* were present in some quantity in all the vascular beds assayed by bulk RNA-seq, *Jak3* appeared as the main Jak family kinase to

contribute to the transcriptomic identity of the bone marrow vasculature in particular (Fig. 1e and Supplementary Fig. 1d). In sum, the centrality and overrepresentation of *Jak3* alongside angiocrine factors in the bone marrow further suggested its involvement in aspects of hematopoietic function.

Although the presence of JAK3 in endothelium has been reported before in human cells[48], most of the research to date has focused on its role in hematopoietic cell biology[43,58]. Analyses of open-source-available murine BMEC RNA-seq datasets reported by another group[26,27] showed that Jak family gene expression decreased to negligible levels in vitro under xenobiotic culture conditions (Fig. 1f). We found an increase in *Jak3* expression in BMECs that were placed in co-culture with hematopoietic cells under xenobiotic-free conditions (Fig. 1g, left). Together, our observations from new and historical, publicly available RNA-seq data indicated that *Jak3* expression is enriched in primary BMECs compared to those of other organs, including the ones rich in sinusoidal endothelium such as the liver. These analyses also suggested that Jak3 function is likely specific to the bone marrow, a primary hematopoietic organ, where it may regulate microenvironmental cues that affect the HSC compartment.

**Jak3 regulates sinusoidal specification of BMECs.** To investigate the organ-specific purview of Jak3, we used a global knock-in mouse model that carries an enzymatically inactivated Jak3 kinase domain (*Jak3*$^{-/-}$; Supplementary Fig. 2)[44]. We performed bulk RNA-seq on primary BMECs from C57BL/6 (wild-type, WT) and *Jak3*$^{-/-}$ mice, respectively, and compared them to their counterparts from the lungs, which, as derived from a non-hematopoietic vascular bed, served as negative controls. PCA of these data showed that WT vs. *Jak3*$^{-/-}$ BMECs accounted for most of the variance across all samples (Fig. 2a). PC1 separated WT BMECs from *Jak3*$^{-/-}$ BMECs and both WT and *Jak3*$^{-/-}$ lung ECs, yielding 73.7% of the variance. PC2 accounted for another 13.3% of the variance and separated *Jak3*$^{-/-}$ BMECs from the rest of the samples. These results indicated that even though *Jak3*$^{-/-}$ mice are devoid of functional Jak3 in all tissues, their vascular defect may be localized to the bone marrow niche and so bring about a hematopoietic phenotype.

Differential expression analyses of *Jak3*$^{-/-}$ against WT ECs yielded 3922 transcripts among bone marrow-derived samples, 13 transcripts among their lung counterparts, and 15 transcripts that were differentially expressed in both cohorts (Fig. 2b). GSEA of the 3937 differentially expressed transcripts in *Jak3*$^{-/-}$ against WT BMECs yielded enrichment of KEGG terms "Hematopoietic cell lineage" and "B cell receptor signaling pathway" (Fig. 2c), suggesting that the loss of *Jak3* may affect the interactions between the ECs assayed and their neighboring blood cells. GSEA also showed dysregulation of signaling pathways with known hematopoietic function, namely the Notch, Tgfβ, Hippo, and Hedgehog signaling pathways[24,59,60]. More specifically, Notch ligands *Jag1*, *Jag2*, and *Dll4*, as well as receptors *Notch1* and *Notch3* and transcription factor *Hes1*, all characteristic of the bone marrow vascular niche and involved in HSC regulation[20,61–64], were decreased in BMECs devoid of enzymatically active *Jak3* (Fig. 2d). Exhaustive mining of the gene lists that contributed to the KEGG terms retrieved by GSEA recovered multiple additional angiocrine factors with pro-angiogenic and -hematopoietic functions that were also downregulated in the absence of *Jak3*, among them are: *Fgf2*[65,66], *Wnt5a*[67,68], *Tgfb3*[69], *Bambi*[64,70,71], *Bmp2*[64,72], and *Bmp4*[64,73] (Fig. 2d and Supplementary Fig. 3a). Taken together, dysregulation of key angiocrine factors and transcription factors indicated that absence of *Jak3* is concomitant with a bone marrow vascular program that is deleterious for HSC maintenance at steady state. Furthermore,

the corresponding transcripts exhibited little to no change between WT and *Jak3*$^{-/-}$ lung ECs, supporting the notion that Jak3, as expressed by the vasculature, is enriched in the bone marrow and an active participant in the regulation of organotypic EC function.

Targeted differential expression analyses showed consistent downregulation of transcripts associated with sinusoidal endothelial identity, such as *Sele*, *Nrp1*, *Stab2*, *Nr2f2*, *Flt4*, *Vcam1*, *Aplnr*, and *Jag2*, in *Jak3*$^{-/-}$ compared to WT BMECs (Supplementary Fig. 3b, left). Transcripts associated with arterial fate like *Cxcr4*, *Vwf*, *Aqp1*, and *Ly6a* (which encodes protein Sca1) were upregulated in the absence of enzymatically active Jak3. By contrast, none of these transcripts were differentially expressed in *Jak3*$^{-/-}$ compared to WT lung endothelial samples (Supplementary Fig. 3b, right). Loss of the gene expression signature characteristic of sinusoids upon its functional deletion suggested that *Jak3* may be associated with sinusoidal fate acquisition or arterial fate suppression. To verify whether *Jak3* expression is heterogeneous within primary WT BMECs at steady state, we combined published single-cell RNA-seq (scRNA-seq) datasets obtained from non-hematopoietic, bone marrow niche cells as reported by two groups[62,74]. The approach by Tikhonova et al. consisted of isolating and sequencing three known constituent populations of the bone marrow niche discretely, using reporter mice to identify *Cdh5*$^+$, *Lepr*$^+$, or *Col2.3*$^+$ cells, respectively. In Baryawno et al., bone marrow cells were isolated either by flushing the marrow or by crushing the whole bone to include osteogenic cells, without further cell-specific selection other than to eliminate hematopoietic cells. We were able to successfully merge and batch-correct these independently generated datasets to render an even more comprehensive map of the bone marrow niche at steady state. Our analysis yielded clusters according to cell identity, not sample provenance (Fig. 2e, top); cells from either dataset contributed to the identified clusters, as expected (Fig. 2e, bottom). As a result, we were able to confirm that *Jak3* is present in the bone marrow microenvironment, especially in ECs (Fig. 2f). Within the EC clusters (*Cdh5*$^+$), *Jak3* expression was increased in the sub-populations corresponding to sinusoidal (*Flt4*$^+$ and *Stab2*$^+$) rather than arterial/arteriolar (*Ly6a*$^+$) cells (Supplementary Fig. 4a).

Further, *Jak3* transcript abundance and expression pattern was similar to those of angiocrine factors such as *Dll4* and *Inhbb* as well as receptors like *Notch1* and *Tgfbr2* (Supplementary Fig. 4b). These transcripts not only contributed to the two KEGG terms retrieved by GSEA that are most suppressed in BMECs in the absence of *Jak3*, they also figure among the severely downregulated genes in the bone marrow vasculature specifically (Fig. 2c, d). Notably, deletion of *Dll4* from the bone marrow vascular niche has been shown to result in premature, non-cell-autonomous myeloid differentiation of HSCs in Tikhonova et al.[62]. In all, through combined transcriptomic analyses, we have shown that *Jak3* is enriched in sinusoidal BMECs and may regulate aspects of sub-organotypic fate acquisition as well as hematopoietic function, since the loss of its enzymatically active form was correlated with both the emergence of arterial marker expression and the downregulation of genes such as *Dll4*.

These findings were corroborated by means of immunofluorescent staining followed by flow cytometry and microscopy. Flow cytometric analyses of CD45$^-$CD31$^+$VE-Cad$^+$ BMECs generally yield CD31 vs. VE-Cad plots that consist of two double-positive sub-populations: a larger, VE-Cad$^{hi}$ sub-population that is also Sca1$^{lo}$ and corresponds to the sinusoids, and a smaller, VE-Cad$^{lo}$ sub-population that is Sca$^{hi}$ and thus arteriolar[26,34] (Supplementary Fig. 5a). *Jak3*$^{-/-}$ BMECs appeared largely devoid of the latter, VE-Cad$^{lo}$Sca1$^{hi}$ sub-population conventionally assigned to arterioles. Since loss of Jak3 had

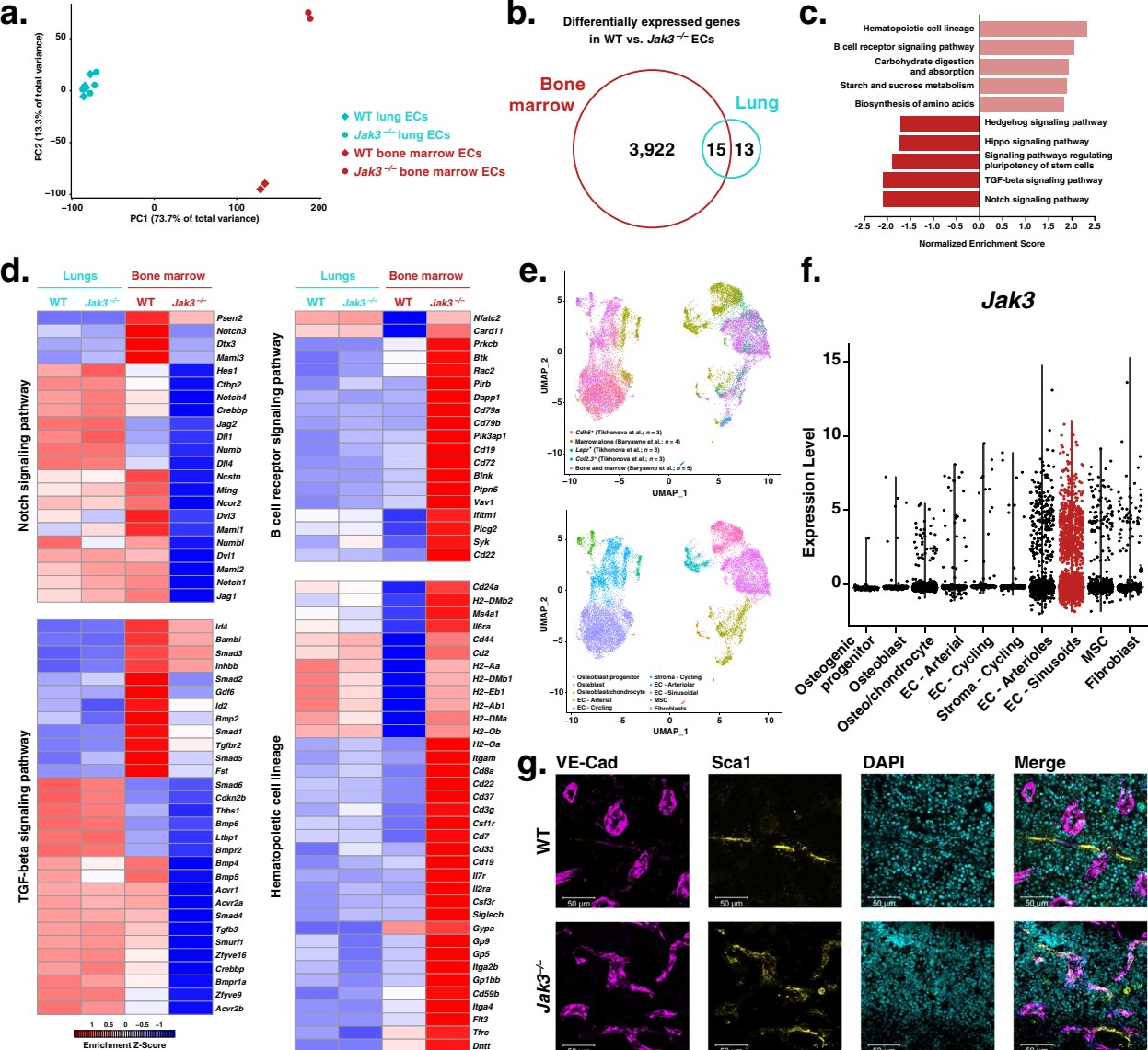

**Fig. 2 Loss of Jak3 alters angiocrine program and canonical endothelial cell (EC) protein expression pattern in the bone marrow. a** Principal component analysis showing principal component 1 (PC1) plotted against PC2 using RNA sequencing (RNA-seq) counts from wild-type (WT) vs. $Jak3^{-/-}$ lung (blue) and bone marrow (red) ECs ($n = 2$ or 3 per group). **b** Venn diagram displaying number of intersecting overrepresented gene transcripts that resulted from differential expression analyses of WT and $Jak3^{-/-}$ bone marrow and lung ECs (FDR < 0.05). **c** Bar plot of enrichment scores of the 10 most dysregulated Kyoto Encyclopedia of Genes and Genomes (KEGG)[54–56] terms generated by gene set enrichment analysis (GSEA) using the $\log_2$ fold change of the differentially expressed gene transcripts in $Jak3^{-/-}$ bone marrow ECs compared to their WT counterparts. **d** Heatmaps depicting normalized transcript abundances of genes that contributed to the 2 most upregulated and the 2 most downregulated KEGG terms by GSEA ($n = 2$ or 3 per organ). **e** Uniform Manifold Approximation and Projection (UMAP) obtained by single-cell RNA sequencing (scRNA-seq) of bone marrow cells. Raw scRNA-seq counts were obtained from Tikhonova et al.[62] (GEO: GSE108892) and Baryawno et al.[74] (GEO: GSE128423), combined, and re-analyzed for this publication. Data are shown clustered by provenance (top) and tissue type (bottom). MSC: mesenchymal stem cells. **f** Violin plot illustrating Jak3 transcript abundance as obtained from combined scRNA-seq analysis of bone marrow cells from Tikhonova et al.[62] (GEO: GSE108892) and Baryawno et al.[74] (GEO: GSE128423). Each dot represents one cell; the cluster identified as sinusoidal endothelial is highlighted in red. **g** Representative anti-VE-Cad (magenta), anti-Sca1 (yellow), and DAPI (cyan) immunofluorescent staining of WT (top) and $Jak3^{-/-}$ (bottom) femurs ($n = 3$). Scale bar is 50 μm.

seemed to promote an arteriolar transcriptional program (Supplementary Fig. 3b, left), we measured Sca1 protein expression to reconcile these seemingly conflicting results. By flow cytometry, the sinusoidal (VE-Cad$^{hi}$) sub-population showed increased Sca1 expression in knockouts compared to controls (Supplementary Fig. 5b). We observed the same results by immunofluorescent staining of the bone marrow, where Jak3$^{-/-}$ mice displayed Sca1$^+$ sinusoids (Fig. 2g), and of other hematopoietic organs, such as the thymus and the spleen

(Supplementary Fig. 6a). Jak3$^{-/-}$ lungs, heart, and brain, however, showed no vascular phenotype by immunofluorescent staining (Supplementary Fig. 6b). Instead of the disappearance of arterioles, our results suggest that bone marrow sinusoids display markers of arterial fate in the absence of $Jak3$. In sum, we observed not only $Jak3$ induction in ECs after hematopoietic co-culture in vitro (Fig. 1g), but also $Jak3$ enrichment in the sinusoidal compartment of the bone marrow in vivo (Fig. 2f and Supplementary Fig. 2). Loss of $Jak3$ disrupted the phenotype of

bone marrow sinusoids at the transcript level and altered protein zonation of canonical arterial marker Sca1. Since HSCs are thought to localize to either the arteriolar[4,38,62,75,76] or the sinusoidal[1,13,36,77] vascular niches in the bone marrow, our data suggested that Jak3 is implicated in some aspect of HSC regulation as well.

**Jak3 regulates LT-HSC maintenance and expansion**. HSC function relies on bone marrow niche chemotaxis effectuated by Kitl, Pgf/Vegfb, and Cxcl12[3,10,78–81] as well as angiocrine signaling via FGF2, IGFBP2, and ANGPT1[24,64] and Notch ligands Jag1, Jag2, Dll4, and Dll1[19,20]. To carry out studies of hematopoietic maintenance and expansion in vitro, our laboratory developed a model of the vascular niche that consists an overexpression of adenovirus 5 early gene 4 (E4ORF1) in human umbilical vein endothelial cells (HUVECs)[82]. HSCs thrive under serum-free conditions that are noxious for EC culture, and E4ORF1 endows HUVECs with the ability to grow in completely xenobiotic-free medium while maintaining their angiocrine repertoire without oncogenic transformation[64]. Having shown that loss of Jak3 function lead to a dysregulated vascular program in the murine bone marrow, we set out to measure whether JAK3 overexpression would enhance LT-HSC expansion. For this purpose, we exogenously expressed JAK3 in our model of the vascular niche (Fig. 3a and Supplementary Fig. 7a, b) and placed the cells in co-culture with murine Lineage⁻cKit⁺Sca1⁺ (LKS) cells for 8 days in the presence of 50 ng μl⁻¹ recombinant Kitl, Tpo, and Flt3, respectively (Fig. 3b). While we observed no difference in the proportion of phenotypic LKS cells after expansion (Fig. 3c), we did measure increased LKS cell output following co-culture with JAK3-overexpressing vascular niche cells compared to HUVEC-E4ORF1 transduced with empty retroviral vectors as controls (Fig. 3d). A kinetics assay carried out throughout the expansion process showed that the number of LKS cells expanded on Vector-transduced ECs began to plateau on day 4, while their counterparts expanded on JAK3-transduced ECs continued to grow through the end of the experiment (Supplementary Fig. 8a). Concurrent cell cycle analyses showed that most LKS cells on either Vector- or JAK3-transduced ECs are likely to be cycling for the duration of the expansion, with a proportion of cells in G1 measurable only on day 8 (Supplementary Fig. 8b). Total CD45⁺ and Lineage⁺ cell expansion, respectively, echoed the results obtained in the LKS sub-population (Supplementary Fig. 8c). These results indicated that, compared to controls, JAK3-overexpressing ECs promote a steadier, more proliferative state in the LKS with which they are co-cultured. Next, CFU assays were performed to assess the progenitor ability of the hematopoietic cells after in vitro co-culture. Control groups yielded significantly increased CFU monocytic (CFU-M) and CFU granulocytic/monocytic (CFU-GM) potential compared to JAK3-enhanced niche cells (Fig. 3e). No significant change in CFU granulocytic (CFU-G) or CFU granulocytic/erythrocytic/macrophagic/megakaryocytic (CFU-GEMM) potential was detected between Vector- and JAK3-enhanced ECs. To test whether JAK3-overexpressing vascular niche cells provided an expansion advantage to LT-HSCs, we performed competitive transplantations of CD45.2⁺ whole bone marrow (WBM) cells and congenic, CD45.1⁺ LKS cells expanded on either control or JAK3-enhanced ECs. Long-term, multilineage engraftment analyses by flow cytometry revealed no change in chimerism or lineage distribution of myeloid (Cd11b⁺ or Gr1⁺) or lymphoid (CD3⁺ or B220⁺) donor cells for 16 weeks (Supplementary Fig. 8d, e). To quantify the frequency of LT-HSCs after co-culture, we designed a limiting-dilution assay (LDA) by which expanded CD45.1⁺ LKS cells were transplanted into lethally irradiated CD45.2⁺ recipients with a

co-infusion of $1 \times 10^6$ CD45.2⁺ WBM cells (Fig. 3f). Endpoint flow cytometric analyses of donor (CD45.1) chimerism were used as input for LDA calculations 20 weeks following transplantation (Fig. 3g). Adhering to the Extreme Limiting Dilution Analysis (ELDA) toolkit[83], we determined the frequency of HSCs after co-culture on JAK3-overexpressing ECs to be 1:92.4 cells, which was equivalent to 3511 LT-HSCs in culture on day 9 (Fig. 3h). This compared to a 1:746 HSC frequency within the control-expanded LKS population, which was equivalent to 253 LT-HSCs in culture at day 9 for that group. We therefore found that the LT-HSC maintenance potential of JAK3-enhanced vascular niche cells was increased almost 14-fold and significant ($p = 7.82 \times 10^{-6}$) with respect to controls. Flow cytometric analyses of multilineage engraftment were run across highest dose transplant recipients and revealed no lineage distribution abnormalities between experimental and control groups other than a discrepancy in the B cell compartment (Supplementary Fig. 8f). To show viability and HSC potential of the grafts subjected to LDA, secondary transplantations were carried out using WBM from highest dose primary-transplant recipients. Flow cytometric analyses of multilineage engraftment were run after 16 weeks, revealing no lineage distribution abnormalities between experimental and control groups (Supplementary Fig. 8g). Altogether, these data show that JAK3 expression improves the LT-HSC maintenance potential of human vascular niche cells in vitro.

We have shown that loss of enzymatically active Jak3 alters the transcriptional program of murine bone marrow, not lung, endothelium (Fig. 2a, b, d and Supplementary Fig. 3a, b). As a primary hematopoietic organ, the bone marrow vascular niche has been shown to display increased LT-HSC maintenance and expansion potential in vitro than its pulmonary counterpart[26,27]. We set out to investigate whether loss of functional Jak3 would disrupt this in vitro phenotype in an organotypic manner as well using a mouse-on-mouse expansion strategy. For that purpose, we placed LKS cells in co-culture with WT and Jak3-depleted lung- and bone marrow-derived ECs. Similar to our mouse-on-human system, LKS cells were expanded under xenobiotic-free conditions and in the presence of 50 ng μl⁻¹ recombinant Kitl, Tpo, and Flt3 for 8 days. We observed a decrease in the hematopoietic expansion potential of lung ECs in the absence of functional Jak3 both proportionally (Fig. 4a) and by absolute cell number (Fig. 4b); however, CFU assays of expanded LKS cells yielded no significant difference between experimental and control groups (Fig. 4c). A kinetics assay carried out throughout the process of expansion on WT and Jak3-depleted lung-derived ECs, respectively, mirrored the expansion results (Supplementary Fig. 9a), while cell cycle analyses showed little to no difference between experimental and control groups (Supplementary Fig. 9b). Unexpectedly, loss of Jak3 brought about contrasting results in BMECs, as we measured an increase in percent and absolute number of immuno-phenotypic LKS cells after expansion on knockout ECs with respect to controls (Fig. 4d, e). As with previous experiments, these results were also corroborated by a kinetics assay (Supplementary Fig. 9c). Concurrent cell cycle analyses showed a sizable proportion of LKS cells on either WT or Jak3⁻/⁻ BMECs in G1 for the duration of the experiment (Supplementary Fig. 9d, left), which not only contrasted our findings from lung EC expansions (Supplementary Fig. 9b, left) but also from hematopoietic co-cultures with human ECs (Supplementary Fig. 8b, left). By day 8, however, we measured about twice as many cycling LKS cells in co-culture with Jak3-depleted BMECs compared to controls (Supplementary Fig. 9d, right). We ascribe these results to an increase in the proliferation of progenitor cells, since the accompanying CFU assays showed the cells expanded onto Jak3-depleted BMECs harbored significantly decreased CFU-GM and CFU-GEMM potentials,

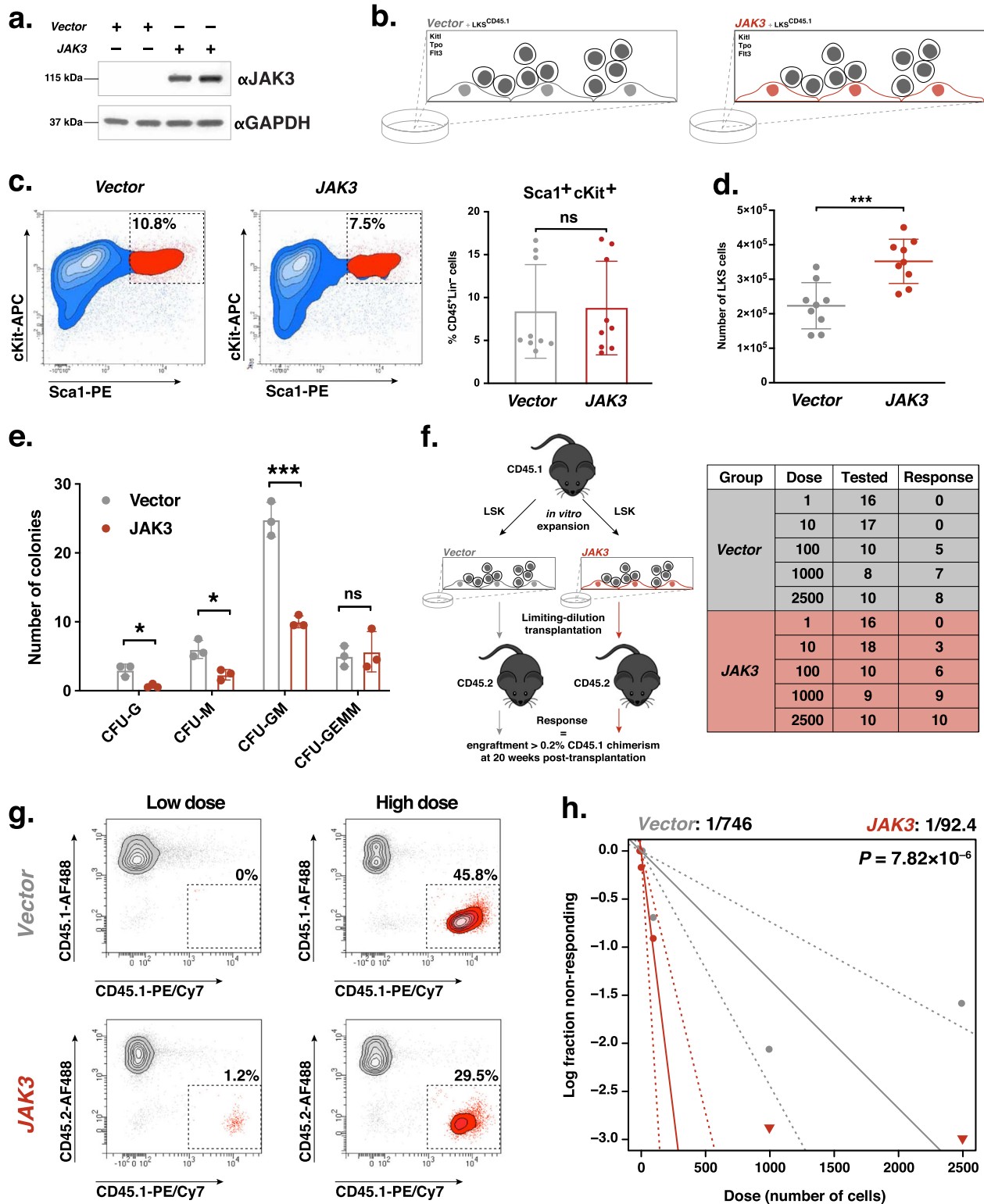

respectively (Fig. 4f). Next, we set out to test whether a *Jak3*−/− bone marrow niche is deleterious for LT-HSC maintenance in vivo. To that end, we designed an LDA that consisted of two rounds of WT WBM (CD45.1+) transplantations, the first into WT or *Jak3*−/− mice, respectively (Fig. 4g, left). After 20 weeks, primary-transplanted mice were euthanized and LKS cells from experimental and control cohorts were transplanted in a limiting-dilution manner into secondary WT recipients (Fig. 4g, right). At the moment of WBM collection for LDA, WT and *Jak3*−/−

primary-transplanted mice displayed no multilineage reconstitution defect in the peripheral blood (PB) (Supplementary Fig. 9e). However, the proportion of LKS cells in the bone marrow was marginally higher for *Jak3*−/− mice (Supplementary Fig. 9f), but only a fifth of these were CD45.1+, which suggested that the rest of the graft had been lost (Supplementary Fig. 9g). By contrast, most of the LKS cells in the bone marrow of WT primary-transplanted mice were CD45.1+. Regardless, we carried out secondary transplants in a limiting-dilution fashion using the

**Fig. 3 JAK3 overexpression in endothelial cells (ECs) provides an expansion advantage to hematopoietic stem cells during in vitro co-culture.**
**a** Western blot of *Vector*- or *JAK3*-transduced human ECs for in vitro co-culture experiments ($n = 2$). **b** Schematic depiction of co-culture setup. CD45.1$^+$Lineage$^-$cKit$^+$Sca1$^+$ (LKS) cells were plated onto confluent monolayers of *Vector*- (gray) or *JAK3*- (red) transduced ECs under serum-free conditions and in the presence of Kitl, Tpo, and Flt3 for 8 days. **c** Representative flow cytometry plots depicting endpoint LKS cell expansion experiment gated on CD45.1$^+$Lineage$^-$ live single cells (left and center; $n = 9$). LKS cell percentages were averaged (right). **d** Total number of LKS cells following expansion as gated on DAPI$^-$CD45.1$^+$Lineage$^-$ live single cells ($n = 9$). Center lines represent the average number of cells. **e** Quantification of colony-forming unit (CFU) assays using expanded LKS cells measuring CFU-granulocytic (CFU-G), CFU-monocytic (CFU-M), CFU-granulocytic/monocytic (CFU-GM), and CFU granulocytic/erythrocytic/macrophagic/megakaryocytic (CFU-GEMM) potential ($n = 3$). **f** Schematic depiction of transplantation setup (left). CD45.1$^+$ LKS cells were expanded onto *Vector*- or *JAK3*-transduced ECs, then transplanted into CD45.2$^+$ recipients in a limiting-dilution manner. Positive engraftment was deemed as 0.2% and higher donor chimerism at experiment endpoint (20 weeks post-transplant. Results summary table (right) where the Group column refers to the cells used in co-culture, the Dose column refers to the number of LKS cells transplanted in each mouse, the Tested column refers to the number of mice that survived to experiment endpoint (week 20), and the Response column refers to the number of engraftment mice at experiment endpoint. **g** Representative flow cytometry plots depicting endpoint chimerism after co-culture with *Vector*- (gray) or *JAK3*- (red) transduced ECs and transplantation into lethally irradiated recipients in a limiting-dilution manner ($n = 5$ or 10 mice per dosage cohort). Plotted events were gated on DAPI$^-$ single cells. Low and high doses correspond to initial injections of 10 and 1000 LKS cells, respectively. **h** Log-fraction plot of limiting dilution analysis showing the frequency of long-term multilineage engraftment following limiting-dilution assay. Dashed lines indicate 95% confidence intervals. Stem cell frequency and statistics were determined using Extreme Limiting Dilution Analysis. ***$p$ value < 0.001, *$p$ value < 0.05, and ns (not significant) > 0.05, by unpaired, two-tailed Student's $t$ test. On all plots, each replicate is represented by an individual dot. All error bars represent the standard deviation from the mean.

same number of CD45.1$^+$ LKS cells in experimental and control cohorts of recipient mice, respectively. Endpoint flow cytometric analyses of donor (CD45.1) chimerism were used as input for LDA calculations 20 weeks after secondary-transplanted (Fig. 4h). We calculated the frequency of LT-HSCs after primary transplantation, homing, and engraftment into the bone marrow of Jak3$^{-/-}$ mice as 1:15,716 cells. This frequency was equivalent to 127 LT-HSCs educated in a *Jak3*-depleted bone marrow niche (Fig. 4i). Within the control-educated WBM population, we measured a 1:239 frequency of functional HSCs, i.e., 8368 LT-HSCs after primary transplantation. We therefore found a 66-fold decrease in LT-HSCs upon in vivo education in a bone marrow niche devoid of enzymatically active Jak3 ($p = 6.8 \times 10^{-7}$). Interestingly, we only detected bone marrow engraftment in one of the 35 secondary transplant recipients from the group subjected to *Jak3*$^{-/-}$ education. Flow cytometric analyses of multilineage engraftment were run across highest dose control-transplant recipients and the single grafted Jak3$^{-/-}$ niche-educated specimen, revealing no lineage distribution abnormalities between the groups (Supplementary Fig. 9h). For this multilineage analysis, we incorporated data from all recipient mice regardless of extent of donor chimerism. Taken together, these results showed that functional Jak3 expression in the bone marrow niche is necessary for LT-HSC maintenance in vivo.

## Discussion

The bone marrow microenvironment is thought to provide an intricate milieu that preserves the repopulating capacity of HSCs or directs their differentiation in a carefully orchestrated manner. We have shown that Jak3, as expressed in the murine bone marrow, is necessary for local arterial-venous zonation and for the maintenance of hematopoietic function. Our approach combined in-house and publicly available RNA sequencing (RNA-seq) obtained in bulk and at the single-cell level with functional studies in vitro and in vivo using a *Jak3*$^{-/-}$ mouse model.

Although the bone marrow is the primary hematopoietic organ in mammalians from the perinatal stage and onward, the liver, a site for HSC expansion in utero[84], possesses vestigial hematopoietic function through adulthood[85,86]. By means of bulk transcriptomic analyses, we contrasted the vascular beds of the bone marrow and the liver as well as those of non-hematopoietic organs such as the lungs, heart, and kidneys. These analyses yielded unique transcriptomic signatures for each of the organs assayed that were characterized by overrepresentation of known markers such as *Gata4*[25] in the liver and *Slc14a1* in the kidneys[52]. In addition, we found that *Jak3* expression was uniquely enriched

in the bone marrow vascular bed. *Jak3* encodes a kinase that has been characterized predominantly in immune cells[58]. Here, we show that *Jak3* is not only enriched in BMECs, but also that its expression in endothelium is modulated by direct contact with hematopoietic cells in vitro.

*Jak3* expression was detected in all the vascular beds assayed, but we found that its endothelial function is localized to the bone marrow. There was no measurable transcriptomic difference between *Jak3*$^{-/-}$ and WT lung ECs, whereas the expression pattern of multiple surface markers, angiocrine factors, and transcription factors was dysregulated in *Jak3*$^{-/-}$ compared to WT BMEC samples. We used open-source single-cell RNA-seq data to further show that *Jak3* expression is enriched in the sinusoidal, as opposed to the arterial, compartment of the bone marrow vasculature. Moreover, *Jak3*$^{-/-}$ BMECs displayed increased expression of known arterial markers such as *Procr*, *Gja5*, and *Efnb2* at the expense of gene transcripts associated with sinusoidal endothelium like *Vcam1*, *Sele*, and *Nrp1*. Anti-Sca1 immunofluorescent staining, routinely used to discriminate arterial from sinusoidal endothelium in the bone marrow, yielded positive results on *Jak3*$^{-/-}$ sinusoids, too. Taken together, our data show that loss of *Jak3* results in aberrant arterial-venous zonation in the bone marrow at both the transcript and protein levels.

Of note, we found that *Jak3*$^{-/-}$ BMECs display decreased expression of *Cxcl12* and *Jag2* (Supplementary Fig. 3b, left), two angiocrine factors that are necessary for proper hematopoietic function[20,63]. Given that *Jak3* is expressed by ECs in a hematopoietic milieu and its presence is associated with the regulation of angiocrine factors in the microenvironment of HSCs, we used an in-house HSC expansion model to gauge the effect of Jak3 on murine hematopoiesis in vitro and in vivo. By means of limiting-dilution transplantation assays, we showed that *JAK3*-overexpression provided an expansion advantage to HSCs in vitro; conversely, the *Jak3*$^{-/-}$ bone marrow microenvironment proved deleterious for long-term HSC maintenance. Consistent with our loss-of-function findings at the transcriptomic level, *Jak3*$^{-/-}$ lung ECs yielded no measurable phenotype following co-culture with hematopoietic cells. These functional readouts implicate the *Jak3*-expressing niche as a novel player in the complex HSC microenvironment of the bone marrow. However, since our work relied on a global knockout mouse, more research will be necessary to definitively ascertain whether sinusoidal ECs are responsible for the observed hematopoietic phenotype. In particular, whether *Jak3* expression by other cells in the bone marrow niche, which has been observed by single-cell RNA-seq[62,74], can affect HSC behavior remains to be shown.

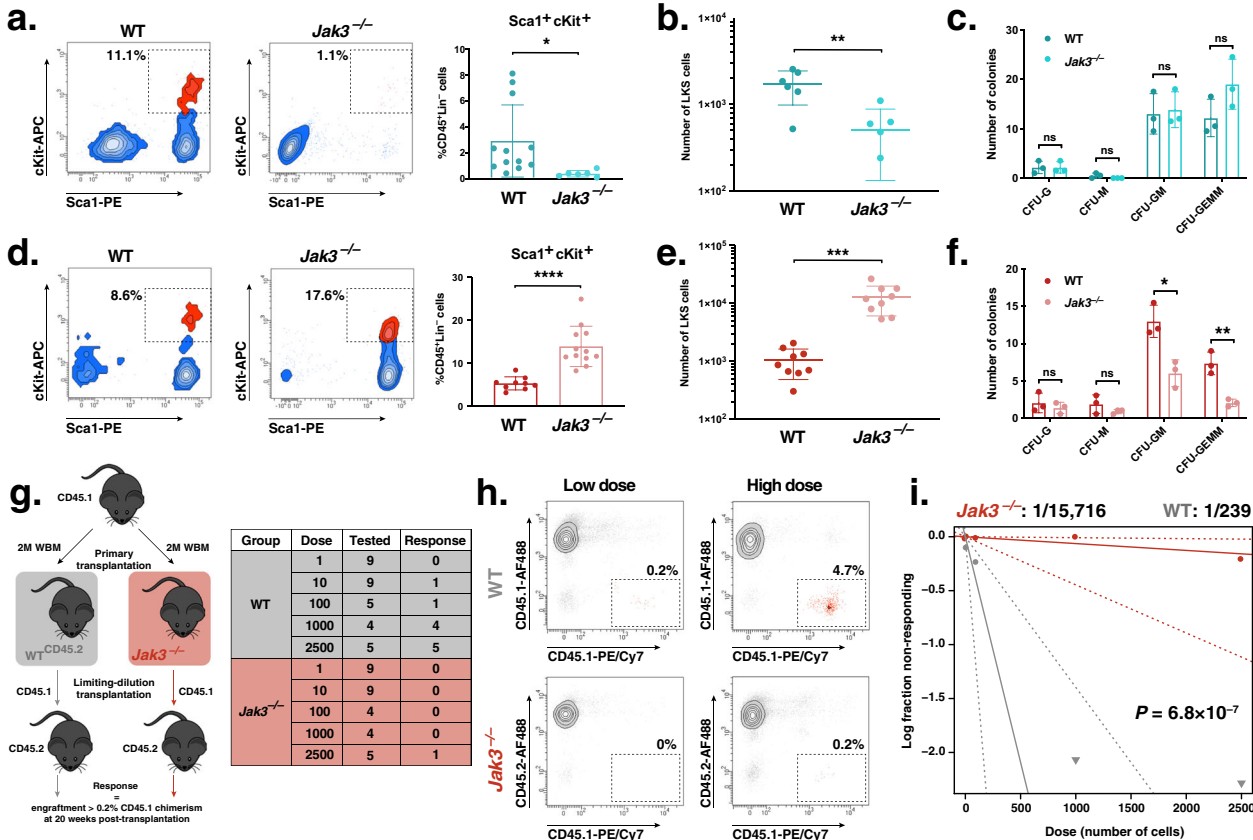

**Fig. 4 Loss of *Jak3* results in a hematopoietic defect in vitro and in vivo. a** Representative flow cytometry plots depicting endpoint Lineage⁻cKit⁺Sca1⁺ cell expansion on wild-type (WT; dark blue) or *Jak3⁻/⁻* (light blue) murine lung endothelial cells (ECs). Plotted events were gated on CD45.1⁺Lineage⁻ single live cells (left and center; $n = 6$). LKS cell percentages were averaged (right). **b** Total number of LKS cells following expansion on lung ECs as gated on DAPI⁻CD45.1⁺Lineage⁻ live single cells ($n = 6$). Center lines represent the average number of cells. **c** Quantification of colony-forming unit (CFU) assays measuring the CFU-granulocytic (CFU-G), CFU-monocytic (CFU-M), CFU-granulocytic/monocytic (CFU-GM), and CFU granulocytic/erythrocytic/macrophagic/megakaryocytic (CFU-GEMM) potentials of LKS cells following expansion onto murine lung WT (dark blue) or *Jak3⁻/⁻* (light blue) ECs ($n = 6$ per group). **d** Representative flow cytometry plots depicting endpoint LKS cell expansion on wild-type (WT; dark red) or *Jak3⁻/⁻* (light red) murine bone marrow ECs (BMECs). Plotted events were gated on DAPI⁻CD45.1⁺Lineage⁻ single cells (left and center; $n = 6$). LKS cell percentages were averaged (right). **e** Total number of LKS cells following expansion on BMECs as gated on DAPI⁻CD45.1⁺Lineage⁻ live single cells ($n = 6$). Center lines represent the average number of cells. **f** Quantification of CFU assays measuring the CFU-G, CFU-M, CFU-GM, and CFU-GEMM potentials of LKS cells following expansion onto murine bone marrow WT (dark red) or *Jak3⁻/⁻* (light red) ECs ($n = 6$ per group). **g** Schematic depiction of transplantation setup. A dose of 2 million (2 M) CD45.1⁺ donor whole bone marrow (WBM) cells was transplanted into WT or *Jak3⁻/⁻* CD45.2⁺ primary recipients, respectively (left). Secondary transplantations were setup using purified, primary-transplant-derived LKS cells delivered in a limiting-dilution manner into WT recipients. Results summary table (right) where the Group column refers to the recipient mice used in primary transplantation, the Dose column refers to the number of LKS cells transplanted in each mouse, the Tested column refers to the number of mice that survived to experiment endpoint (week 20), and the Response column refers to the number of engraftment mice at experiment endpoint. **h** Representative flow cytometry plots depicting endpoint chimerism after in vivo education in a WT (gray) or *Jak3⁻/⁻* (red) bone marrow microenvironment (left; $n = 5$ or 10 mice per dosage cohort). Plotted events were gated on DAPI⁻ single cells. Low and high doses correspond to initial injections of 10 and 1000 LKS cells, respectively. **i** Log-fraction plot of limiting dilution analysis showing the frequency of long-term multilineage engraftment following limiting-dilution assay. Dashed lines indicate 95% confidence intervals. Stem cell frequency and statistics were determined using Extreme Limiting Dilution Analysis. ****$p$ value < 0.0001, ***$p$ value < 0.001, **$p$ value < 0.01, *$p$ value < 0.05, and ns (not significant) > 0.05, by unpaired, two-tailed Student's $t$ test. On all plots, each replicate is represented by an individual dot. All error bars represent the standard deviation from the mean.

The vasculature within each organ is heterogeneous and responds to the specific physiological requirements of its milieu. Here, we show that *Jak3* expression in ECs is not only induced by hematopoietic co-culture in vitro, but also enriched in the bone marrow sinusoidal compartment in vivo. Most notably, we demonstrate that non-cell-autonomous *Jak3* is required for hematopoietic function. Further studies will be necessary to elucidate how Jak3 signals in endothelial compared to immune cells, what cytokines lead to its phosphorylation, and what its downstream targets are. As the only non-promiscuous Jak family kinase, targeting Jak3 in the niche has the potential to become a viable means to manipulate hematopoietic output in myriad contexts, from cancer to injury to congenital disease.

## Methods

**Animals.** Primary ECs from the liver, lungs, heart, kidneys, and bone marrow, respectively, were isolated from WT mice (C57BL/6J; Jackson Laboratory, strain 000664) for RNA sequencing (RNA-seq) or in vitro culture. For loss-of-function experiments, we used a knock-in mouse model (B6;129S4-*Jak3^tm1Ljb*/J; Jackson Laboratory, strain 002852), where the sequence encoding sub-domains 1 through 4 of the *Jak3* kinase domain were replaced by a neomycin resistance cassette, inactivating Jak3 enzymatic activity[44]. ECs from functionally knockout mice (*Jak3⁻/⁻*) were compared to those from WT mice. For transplantation experiments, we

**Table 1 Fluorophore-conjugated antibodies used in endothelial cell isolation and peripheral blood phenotyping FACS experiments.**

| Antigen | Fluorophore | Clone | Cat. no. |
|---|---|---|---|
| VE-Cadherin | AF647 | BV13 | 138006 |
| CD31 | PE/Cy7 | 390 | 102418 |
| CD45 | PE, APC/Cy7 | 30-F11 | 103106, 103116 |
| c-Kit | APC | 2B8 | 105812 |
| Sca1 | PE, PE/Cy7 | D7 | 108108, 108114 |
| Lineage | BV421, FITC | 145-2c11, RB6-8C5, RA3-6B2, Ter-119, M1/70 | 133311, 133302 |
| Ter119 | BV421, APC | Ter119 | 116233, 116212 |
| CD45.2 | AF488, APC/Cy7, AF700 | 104 | 103239, 109824, 109822 |
| CD45.1 | PE/Cy7 | A20 | 110730 |
| CD3 | BV421 | 17A2 | 100228 |
| B220 | APC/Cy7, BV421 | RA3-6B2 | 103239, 103239 |
| Gr1 | PE, FITC | RB6-8C5 | 108408, 103239 |
| CD11b | PE, PE/Cy7, AF700 | M1/70 | 103239, 103239, 101222 |

**Table 2 Mice used for scRNA-seq of murine bone marrow cells in Tikhonova et al.[62] and Baryawno et al.[74].**

| Accession | Name | Mouse strain | Publication | Cell identity |
|---|---|---|---|---|
| GSM2915420 | Col2.3_1 | *Col1a1-cre[ERT2;98]* | Tikhonova | Col1a1+ |
| GSM2915421 | Col2.3_2 | *tdTomato[lox/stop/lox] 99* | et al. (2019)[62] | (osteoblastic) |
| GSM2915422 | Col2.3_3 | | | |
| GSM2915423 | Col2.3_4 | | | |
| GSM2915424 | Lepr_1 | *Lepr-cre;[100]* | | Lepr+ (stromal) |
| GSM2915425 | Lepr_2 | *tdTomato[lox/stop/lox] 99* | | |
| GSM2915426 | Lepr_3 | | | |
| GSM2915427 | Lepr_4 | | | |
| GSM2915428 | Vecad_1 | *Cdh5-cre;[101]* | | VE-Cad+ |
| GSM2915429 | Vecad_2 | *tdTomato[lox/stop/lox] 99* | | (endothelial) |
| GSM2915430 | Vecad_3 | | | |
| GSM2915431 | Vecad_4 | | | |
| GSM3674224 | std1 | C57BL/6 | Baryawno et al. | Whole bone and |
| GSM3674225 | std2 | | (2019)[74] | bone marrow |
| GSM3674226 | std3 | | | |
| GSM3674227 | std4 | | | |
| GSM3674228 | std5 | | | |
| GSM3674229 | std6 | | | |
| GSM3674243 | bm1 | | | Whole bone |
| GSM3674244 | bm2 | | | marrow alone |
| GSM3674245 | bm3 | | | |
| GSM3674246 | bm4 | | | |

isolated and expanded LKS cells from mice that express the CD45.1 isoform of the *Ptprc* gene only (B6.SJL-*Ptprca Pepcb/BoyJ*; Jackson Laboratory, strain 002014). Expanded CD45.1+ LKS cells were transplanted into WT or *Jak3[-/-]* recipients, both exclusively CD45.2+, according to experimental setup. All animal manipulations were carried out with the approval of Weill Cornell Medicine Institutional Animal Care and Use Committee.

**Murine EC isolation and fluorescence-activated cell sorting (FACS).** The livers, lungs, kidneys, hearts, and bones of individual mice were harvested 15 min after retro-orbital injection of 25 μg anti-VE-Cadherin-AF647. Soft tissues were processed as described elsewhere[87]. Briefly, organs were minced and placed in 2.5 mg ml⁻¹ Collagenase A (Roche, 11088793001), 1 U ml⁻¹ Dispase II (Roche, 04942078001), and 50 μg ml⁻¹ DNase I (Roche, 10104159001) in Hank's Balanced Salt Solution (Corning, 21-020-CV) at 37 °C for 20–30 min to create a single-cell suspension. For bone marrow isolation, the sternum, pelvic bones, femurs, and tibias were mechanically denuded of muscle and connective tissue, crushed using a mortar and pestle, and placed in the same digestion solution as above and at the same temperature for 15 min. All tissue digests were filtered through a 40-μm sieve and centrifuged at 500*g* for 10 min. To prevent non-specific antibody binding, pellets were resuspended in a 1:50 solution of FcR Blocking Reagent (Miltenyi Biotec, 130-092-575) in PBS containing 2 mM EDTA (Thermo Fisher Scientific, 15575020) and 0.2% wt/vol bovine serum albumin (BSA; Gemini Bio, 700-100 P), henceforth referred to as blocking buffer, for 10 min at 4 °C. Blocked cell suspensions were co-stained for 30 min at 4 °C using fluorochrome-conjugated antibodies against Ter119, CD45, and CD31 at a concentration of 0.2 μg per 10⁶ cells. Stained samples were washed once and resuspended in blocking buffer with 1 μg ml⁻¹ DAPI (Biolegend, 422801) for viability discrimination. Between 10⁴ and 10⁵ live Ter119⁻CD45⁻CD31⁺VE-Cad⁺ ECs per sample were sorted using a BD FACSAria II instrument and BD FACSDiva 8.0.1 software (BD Biosciences). Respectively, the liver, lungs, heart, and kidneys of an individual mouse corresponded to a single biological replicate; the bones of three mice were combined prior to sorting to obtain enough ECs for a single biological replicate. Gating was determined using unstained controls and fluorescence-minus-one strategies. FACS antibodies were obtained from Biolegend and are specified in Table 1.

**Bulk RNA sequencing analyses.** Sorted EC samples that yielded at least 100 ng total RNA by phenol-chloroform separation using TRIzol LS Reagent (Thermo Fisher Scientific, 10296028) were purified using an RNeasy Mini Kit (Qiagen, 74004), following the manufacturers' instructions. RNA quality was verified using a 2100 Bioanalyzer (Agilent Technologies). RNA library preps were prepared and multiplexed using a TruSeq RNA Library Preparation Kit v2 (non-stranded and poly-A selection; Illumina), after which 10 nM cDNA were subjected to high-throughput sequencing using a HiSeq 4000 (Illumina). The resulting 50-bp-long paired-end reads were checked for quality (FastQC v0.11.5) and processed using the Digital Expression Explorer 2 (DEE2)[88] workflow. Adapter trimming was performed with Skewer (v0.2.2)[89]. Further quality control was done with Minion, part of the Kraken package[90]. Filtered reads were mapped to the mouse reference genome GRCm38 using STAR aligner[91] and gene-wise expression counts were generated using the "-quantMode GeneCounts" parameter. The R package edgeR[92] was used to calculate FPKM and log2 counts per million (CPM) matrices as well as

perform differential expression analyses (Supplementary Data 1). Principal component analyses were performed using log2 CPM values. Data from Poulos et al.[26] were accessed from GEO: GSE61636 and analyzed as above. For GSEA[53] against the KEGG database[54–56], we used WebGestalt[93]. Network visualization was carried out using the STRING app[57] on Cytoscape software v3.8.0[94]. Heatmaps were generated with the pheatmap package in R. Bar plots were generated using Graphpad Prism software v7.0d.

**Single-cell RNA sequencing analyses.** Raw data from Tikhonova et al. (GEO: GSE108892)[62] and Baryawno et al. (GEO: GSE128423)[74] were downloaded from NCBI (Table 2). For both datasets, libraries had been prepared independently using Chromium Single Cell 30 v2 Reagent Kits (10X Genomics). All single-cell analyses were performed using the Seurat version 3.1.4 package in R version 3.6.1. After initial quality control, cells from GSE128423 that were included in the analysis were required to have a minimum 500 genes expressed and a maximum 5500 genes expressed, in addition to a maximum 40,000 UMIs and maximum mitochondrial content of 15%. Similarly, cells from GSE108892 that were included in the analysis were required to have a minimum 1000 genes expressed and a maximum 5000 genes expressed, in addition to a maximum 40,000 UMIs and maximum mitochondrial content of 10%. This resulted in a total 23,020 cells passing quality filters across the 22 samples. Following best practices in the package suggestions we used the sctransform function from Seurat to perform normalization and batch correction between the two datasets. PCA was subsequently performed on the combined set of cell transcripts, and after reviewing principal component heatmaps and jackstraw plots Uniform Manifold Approximation and Projection (UMAP) plots were generated using the top 50 dimensions. Clustering was performed using the Seurat function FindClusters and clustering resolution was set at 0.2. Differential gene expression for gene marker discovery across the clusters was performed using the Wilcoxon rank sum test from the Seurat package.

**Immunohistochemistry.** Organs were harvested 15 min after retro-orbital injection of 25 μg anti-VE-Cadherin-AF647, fixed in 4% paraformaldehyde in PBS overnight at 4 °C, and dehydrated in 30% wt/vol sucrose in PBS for 48 h at 4 °C. Prior to dehydration step, fixed femurs were decalcified in 10% vol/vol EDTA (pH 7.0) in PBS for 72 h at room temperature. For immunofluorescent imaging, dehydrated tissues were snap-frozen in Tissue-Tek O.C.T. Compound (Sakura Finetek, 4583). Tissue sections (15–50 μm) were blocked in 5% vol/vol donkey serum, 0.2% vol/vol Triton X-100 in PBS for 30 min at room temperature and incubated with purified anti-Sca1 (bones; 1:100) or anti-Sca1-PE (soft tissues; 1:200) overnight at 4 °C. After three 5-min washes in 0.2% vol/vol TWEEN 20 (Sigma, P1379) in PBS, bone samples were stained with a donkey-raised, species-matched AF-488 secondary antibody (Life Technologies, A21208; 1:1000) for 1 h at room temperature, then washed again three times. Soft tissue samples were washed, also as above, after overnight staining with conjugated antibody, then mounted. Both bone and soft tissue samples were mounted in Fluoroshield with DAPI (Sigma, F6047). Different preparations were carried out for bones and soft tissues, respectively, since Sca1+ hematopoietic cells are abundant in the former group, thus requiring a more rigorous staining procedure in two steps. For chromogen-based immunohistochemistry, decalcified bones were placed in 40% vol/vol ethanol. Paraffin embedding, tissue sectioning, and anti-Jak3 staining (Abcam, ab203611) of WT and Jak3[-/-] specimens was performed by Histoserv, Inc. Following immunohistochemistry, all image acquisition and image manipulation were carried out, respectively, on an LSM 710 META using ZEN software (both Zeiss).

**Cell culture media**. Human EC medium was prepared by combining and filtering M199 (Sigma, M4530), 10% vol/vol FBS (Omega Scientific, FB07), 50 µg ml$^{-1}$ endothelial mitogen (Alfa Aesar J65416), 100 µg ml$^{-1}$ heparin (Sigma, H3393), 20 mM HEPES (Invitrogen, 15630080), and 1× GlutaMAX (Gibco, 35050-061). Serum-free EC medium was prepared by combining and filtering X-Vivo serum-free medium (Lonza, 04448Q), 100 µg ml$^{-1}$ heparin, 20 mM HEPES, and 1× GlutaMAX. Mouse endothelial cell (EC) medium was prepared by combining and filtering DMEM/Ham's F-12 (Sigma, D6421), 10% vol/vol FBS, 50 µg ml$^{-1}$ endothelial mitogen, 100 µg ml$^{-1}$ heparin, 20 mM HEPES, 1× GlutaMAX, and 1× MEM nonessential amino acids (Corning, 25-025-CI). SB431542 (R&D, 1614) was added to the filtrate at a 5 µM concentration. Hematopoietic stem and progenitor cell (HSPC) expansion medium was prepared by combining and filtering StemSpan SFEM (StemCell Technologies, 09650), 10% vol/vol KnockOut Serum Replacement (Invitrogen, 10828028), 100 µg ml$^{-1}$ heparin, 20 mM HEPES, and 1× GlutaMAX. All media were filtered using a 45-µm filter bottle and stored at 4 °C for up to 2 weeks, with the exception of HSPC expansion medium, which, upon filtering, was distributed into 50-ml conical tubes for storage at –20 °C for up to 6 months. As needed, tubes were thawed in the dark at 4 °C the day before use and 5 ng ml$^{-1}$ human FGF-basic (Peprotech, 100-18), 50 ng ml$^{-1}$ human SCF (Peprotech, 300-07), 50 ng ml$^{-1}$ human TPO (Peprotech, 300-18), and 50 ng ml$^{-1}$ human FLT3 ligand (Peprotech, 300-19) were added. Thawed HSPC expansion medium with cytokines was stored at 4 °C for up to 1 week.

**Human EC isolation and generation of *JAK3*-overexpressing vascular niche cells**. HUVECs were isolated following the method first described in Rafii et al.[95] to produce *E4ORF1*-ECs (E4ECs) as outlined in the work of Seandel et al. and others[19,22,82], with the modifications that appear in Barcia Durán et al.[87]. Briefly, HUVECs were isolated and cultured in human EC medium until transduction with a lentiviral vector encoding the E4ORF1 protein from human adenovirus 5. Exogenous expression of *E4ORF1* confers the capacity to survive under xenobiotic-free conditions to ECs exclusively, so newly transduced E4ECs were positively selected by exposure to serum-free EC medium for 10 days. Selected E4ECs were used in all the expansion experiments described herein after transduction with a retroviral vector encoding *JAK3* or an empty retroviral vector with the same backbone, respectively. Confluent populations of *JAK3*- or *Vector*-transduced lines were contact-inhibited, non-transformed, and could be cultured until their Hayflick limit, generally 20–25 passages from harvest at 37 °C, 5% CO$_2$, and 20% O$_2$.

**JAK3 retrovirus design, production, and transduction**. The human *JAK3* construct was provided by Giorgio Inghirami (Weill Cornell Medicine) and cloned as previously described[96]. Briefly, *JAK3* cDNA (NM_000215) was sub-cloned first into the pcDNA3 expression vector and later into a Pallino retroviral vector that includes an eGFP cassette. Empty retroviral constructs were used as negative controls of transduction, referred to as *Vector* controls. We generated *JAK3* and *Vector* retroviral particles as described in Zamo et al.[97] with the following modifications. Sub-confluent GP2-293 cells (Clontech, 631458) were co-transfected with 1 µg Pallino-based retroviral vector and 1 µg VSV-G, passage number 4–9. Supernatants were collected only once, at 48 h after transfection; they were then filtered, concentrated, and stored at –80 °C for up to 1 year. To assess the quality of virus produced, we added 1, 5, or 10 µl of viral concentrate to 100,000 HUVECs, respectively; assessed eGFP fluorescence by flow cytometry; and used the smallest dose that yielded over 80% fluorescent cells. Consistently, an ~60% confluent well of a 6-well plate of E4ECs transduced with 5 µl of viral concentrate and 5 µg of polybrene (Sigma, TR-1003) in 1 ml of human EC medium yielded over 99% eGFP$^+$ cells. Cells and retroviruses were routinely cultured undisturbed for 72 h.

**Western blot**. *Vector*- or *JAK3*-transduced human E4ECs were lysed using 100 µl of loading buffer containing 50 mM Tris-HCl, pH 6.8 (Thermo Fisher Scientific, RGE3355), 5% vol/vol β-mercaptoethanol (Sigma, M6250), 2% vol/vol sodium dodecyl sulfate, 10% (SDS; Bio-Rad, 1610416), 0.01% vol/vol bromophenol blue (Sigma, 318744), and 10% vol/vol glycerol (Sigma, G5516) in dH$_2$O. Cell lysates were sonicated in 2 intervals of 30 s using a Bioruptor (Diagenode) on the high setting. Proteins (in 10 µl of lysate) were solved by SDS polyacrylamide gel electrophoresis using a NuPAGE 4–12% Bis-Tris Gel (Thermo Fisher Scientific, NP0321PK2). Gels were transferred to nitrocellulose membranes (Bio-Rad, 1620112) blocked in 5% wt/vol milk (Quality Biological, A614-1003) in phosphate buffered saline (PBS; Corning, 21-031-CV), and immuno-blotted with antibodies obtained from Cell Signaling against JAK3 (8827s) or GAPDH (5174), respectively. Primary antibody incubations were carried out in 5% wt/vol reconstituted powder milk (Quality Biological, A614-1003) in PBS. HRP-conjugated secondary goat anti-rabbit antibodies (Jackson ImmunoResearch, 111-035-144) as well as ECL Prime Western Blotting System (GE Healthcare, RPN2232) were used to bind proteins of choice, respectively. Chemiluminescent signals were captured onto X-films (Denville Scientific, E3218).

**Murine EC culture in vitro**. Murine lung and bone marrow ECs were isolated following the method first described in Kobayashi et al.[64], with the modifications that appear in Barcia Durán et al.[87]. Briefly, the lungs and, separately, the sternum, pelvic bones, femurs, and tibias of individual WT or *Jak3$^{-/-}$* mice were excised and

processed as described for FACS previously. ECs were purified using microbeads (Thermo Fisher Scientific, 11035), conjugated to a monoclonal CD31 antibody (Biolegend, 102502), and plated in mouse EC medium. Prior to bead purification, bones were also subjected to Lineage depletion (Miltenyi Biotec, 130-090-858). The resulting lung and bone marrow ECs were transduced with a myristoylated-Akt1-expressing lentivirus[64], after which they remained contact-inhibited and non-transformed, and could be cultured until their Hayflick limit, generally 20–25 passages from harvest at 37 °C, 5% CO$_2$, and 5% O$_2$.

**Murine HSPC expansion in vitro**. For bone marrow isolation, the sternum, pelvic bones, femurs, and tibias of individual CD45.1$^+$ mice were mechanically denuded of muscle and connective tissue, crushed using a mortar and pestle, and filtered through a 40-µm sieve using up to 40-ml blocking buffer. Lineage-committed hematopoietic cells were depleted by magnetic separation using microbeads conjugated to monoclonal antibodies against CD5, B220, CD11b, Gr-1, 7-4, and Ter-119 to bind T lymphocytes, B lymphocytes, monocytes, granulocytes, macrophages, and red blood cells, respectively (Miltenyi Biotech, 130-110-470), following the manufacturer's instructions. To prevent non-specific antibody binding, lineage-depleted cells were centrifuged at 500g for 10 min and resuspended in a 1:50 solution of FcR Blocking Reagent in blocking buffer for 10 min at 4 °C. Blocked cell suspensions were co-stained for 30 min at 4 °C using fluorochrome-conjugated antibodies against CD45, Sca1, cKit, and a Lineage Cocktail at a concentration of 0.2 µg per 10$^6$ cells. Stained samples were washed once and resuspended in blocking buffer with 1 µg ml$^{-1}$ DAPI for viability discrimination. Live CD45$^+$LKS cells were sorted using a BD FACSAria II instrument and BD FACSDiva 8.0.1 software and plated onto confluent monolayers of human (*Vector*- or *JAK3*-transduced) or murine lung or bone marrow (WT or *Jak3$^{-/-}$*) ECs. Cells were co-cultured at 10$^3$ LKS cells per well of a 6-well plate in HSPC expansion medium at 37 °C, 5% CO$_2$, and 5% O$_2$. Expanded HSPC cultures were harvested for flow cytometric analyses and downstream functional assays after 8 days.

**Proliferation and cell cycle studies**. Total cell counts were performed using a Thermo Fisher Scientific Countess II FL (AMQAF1000) and cell counting chamber slides (C10283) following the manufacturer's instructions on days 1, 2, 4, and 8 after expansion setup. Flow cytometry was used to back-calculate LKS cell numbers specifically and to perform cell cycle analyses by combining a PE Mouse Anti-Ki-67 Set (BD Biosciences, 556027) with our LKS staining protocol. Briefly, cells were fixed in 80% vol/vol methanol in PBS at each time point, then washed in blocking buffer at 250 g for 10 min, and resuspended in a 1:50 solution of FcR Blocking Reagent in blocking buffer for 10 min at 4 °C to prevent non-specific antibody binding. Blocked cell suspensions were co-stained for 30 min at 4 °C using fluorochrome-conjugated antibodies against Ki67, CD45, Sca1, cKit, and a Lineage Cocktail at a concentration of 0.2 µg per 10$^6$ cells. Stained cell suspensions were washed once and resuspended in blocking buffer with 1 µg ml$^{-1}$ DAPI in order to fully ascertain cell cycle phase using a BD FACSAria II instrument and BD FACSDiva 8.0.1.

**CFU assays**. CFU assays were setup using murine HSPCs that underwent in vitro expansion on human (*Vector*- or *JAK3*-transduced) or murine lung or bone marrow (WT or *Jak3$^{-/-}$*) ECs as outlined above. LKS cells from control and experimental groups were FACS-purified, suspended in cytokine-rich methylcellulose (StemCell Technologies, 03444), and placed on meniscus-free 6-well plates (StemCell Technologies, 27370) following the manufacturer's instructions. Each of two technical replicates received 100 expanded LKS cells. CFU-granulocytic (CFU-G), -monocytic (CFU-M), -granulocytic/monocytic (CFU-GM), and -granulocytic/erythrocytic/macrophagic/megakaryocytic (CFU-GEMM) capacity was individually and manually scored using the Olympus Stereo Microscope ZSX16 (Olympus America) 14 days after setup. Imaging was performed using the STEMvision instrument (StemCell Technologies). Prior to quantification and imaging, all CFU assays were maintained in a humidified incubator at 37 °C, 5% CO$_2$, and 5% O$_2$.

**Bone marrow repopulation assays**. Competitive transplantations were carried out to compare the long-term, multilineage engraftment potentials of LKS cells following expansion on *Vector*- vs. *JAK3*-transduced ECs. For this purpose, 3000 CD45.1$^+$ expanded LKS cells were delivered retro-orbitally into WT recipients alongside 500,000 WBM WT cells. PB chimerism and lineage distribution of myeloid (Cd11b$^+$ or Gr1$^+$) or lymphoid (CD3$^+$ or B220$^+$) donor cells were assayed weekly for 16 weeks. Limiting-dilution transplantations were carried out to quantify the frequency of multilineage, LT-HSCs after (i) in vitro expansion on human (*Vector*- or *JAK3*-transduced) ECs or (ii) in vivo education in a WT or *Jak3$^{-/-}$* bone marrow microenvironment. For (i), expanded LKS cells from control and experimental groups were FACS-purified and delivered retro-orbitally in a limiting-dilution fashion into recipient mice. Control and experimental groups, respectively, were allotted 35 mice divided into five cohorts corresponding to the following dosage system: 1, 10, 100, 1000, and 2500 sorted LKS cells post-expansion. The two lowest doses of 1 and 10 cells were allotted 10 mice each, whereas 5 mice were allotted to the remaining three dosage cohorts of 100, 1000, and 2500 cells. For (ii), a primary transplant of 2 million CD45.2$^+$ WBM cells were delivered retro-orbitally into 5 WT and *Jak3$^{-/-}$* mice. At 20 weeks post-

transplantation, donor-derived LKS cells were FACS-purified as described previously and delivered retro-orbitally into secondary recipient mice using the same limiting-dilution scheme as in (i). All limiting-dilution transplant recipients in (i) and (ii) were given a radioprotective dose of 1 million CD45.2$^+$ WBM cells in the same injection. Long-term engraftment was assessed by flow cytometry 20 weeks after limiting-dilution transplantation. HSC frequency and 95% confidence intervals were determined using the ELDA online tool[83] and quantifying only the surviving mice. A secondary transplantation of WBM from highest dose, primary-transplanted mice in (i) was carried out to ensure self-renewal potential of the grafts. This experiment was assayed at 16 weeks and was not necessary for the LDA in (ii), which itself took place as a secondary transplant. All transplants were carried out in mice that had received a lethal dose (950 cGy) of radiation using a cesium-137 irradiator 18 h prior to transplantation.

**Flow cytometric analyses of PB**. Using heparinized micro-hematocrit capillary tubes (Thomas Scientific, 41B2501), ~35 µl PB were retrieved retro-orbitally from anesthetized transplant recipient mice. Red blood cells were depleted using a lysis buffer (Biolegend, 420301) and following the manufacturer's instructions. Lysed cells were washed twice at 300g for 10 min using blocking buffer. After the second wash, and to prevent non-specific antibody binding, lysed cells were resuspended in a 1:50 solution of FcR Blocking Reagent in blocking buffer for 10 min at 4 °C. Blocked cell suspensions were co-stained for 30 min at 4 °C using fluorochrome-conjugated antibodies against Ter119, CD45.2, and CD45.1 at a concentration of 0.2 µg per 10$^6$ cells. Stained samples were washed in blocking buffer and fixed in 1% paraformaldehyde (Agar Scientific, AGR1026) in blocking buffer with 1 µg ml$^{-1}$ DAPI for viability discrimination. Using a SORP-LSR2 instrument and BD FACSDiva 8.0.1 software (BD Biosciences), engraftment was determined when >1% or >0.1% live cells appeared CD45.2$^-$CD45.1$^+$ in limiting-dilution transplantation experiments (i) and (ii), respectively. Highest dose experiments were also co-stained with fluorochrome-conjugated antibodies against CD3, B220, Cd11b, and Gr1 at a concentration of 0.2 µg per 10$^6$ cells. Gating was determined using unstained controls and fluorescence-minus-one strategies. FACS antibodies were obtained from Biolegend and are specified in Table 1.

**Statistics and reproducibility**. All data in bar plots are presented as mean ± s.d. To identify statistical significance between single comparisons, groups of data were compared using an unpaired, two-tailed paired Student's $t$ test. Multiple comparisons were analyzed by one- or two-way ANOVA, as indicated. $p$ values > 0.05 were considered not significant. One asterisk (*) was used to indicate $p$ values < 0.05, two asterisks (**) for $p$ values < 0.01, three asterisks (***) for $p$ values < 0.001, and four asterisks (****) for $p$ values < 0.0001. In all the figures, $n$ refers to the number of biological replicates and is indicated in every figure legend. All statistical analyses and data wrangling were performed using Graphpad Prism software v7.0d. No statistical methods were used to predetermine sample size; all animal experiments included at least five animals per group, and all groups included a litter-dependent, near-balanced ratio of males-to-females. All animals used were 10–12 weeks of age and age- and sex-matched between control and experimental groups when applicable. No animals were excluded from quantification. In addition, transplanted animals were not individually labeled. Nevertheless, transplantation experiments were not randomized, and the investigators were not blinded to allocation of control vs. experimental groups during outcome assessment.

**Reporting summary**. Further information on research design is available in the Nature Research Reporting Summary linked to this article.

## Data availability

Bulk RNA sequencing of primary endothelial cells in Figs. 1 and 2 is available for download from the Gene Expression Omnibus (GEO) repository: GSE159755, hosted by the National Center for Biotechnology Information. Bulk RNA sequencing of in vitro studies by Poulos et al.[26] can be accessed from GEO: GSE61636. Single-cell RNA sequencing from Tikhonova et al.[62] and Baryawno et al.[74] can be accessed from GEOs GSE108892 and GSE128423, respectively. Processed bulk RNA sequencing data are provided in Supplementary Data 1 and 2. Source data for main and supplementary figures are provided in Supplementary Data 3. All primary and supplementary data may be provided by the corresponding author upon reasonable request.

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

## Acknowledgements

We thank María Hernández for the administrative support. We thank G. Inghirami at the Department of Pathology and Laboratory Medicine at Weill Cornell Medicine for his generosity in lending us the *Vector* and *JAK3* retroviral plasmids. J.G.B.D., T.L., F.G., R.L., and S.R. are supported by the Ansary Stem Cell Institute at Weill Cornell Medicine and the Tri-Institutional Stem Cell Initiative (TRI-SCI 2013-032, 2014-023, 2016-013, and 2019-029), a research collaboration between Memorial Sloan Kettering Cancer Center, The Rockefeller University, and Weill Cornell Medicine. S.R. is also supported by grants from the National Institutes of Health (NIH; R35 HL150809, R01s DK095039, HL119872, HL128158, HL115128, HL139056, RC2 DK114777, and U01AI138329); grants from New York State Stem Cell Science (NYSTEM; C026878, C028117, C029156, and C030160); the Weill Cornell Medicine Daedalus Fund for Innovation; the Empire State Stem Cell Board; and the Qatar National Priorities Research Program (NPRP 8-1898-3-392).

## Author contributions

J.G.B.D., S.R., and R.L. designed the study. J.G.B.D. performed endothelial cell isolation and hematopoietic cell isolation, hematopoietic cell expansion and transplantation, and molecular biology assays. T.L. carried out cell culture and imaging experiments. F.G. performed western blots. J.X. performed RNA sequencing. J.G.B.D. performed sequencing analyses with supervision by D.R. and S.H.; J.G.B.D., R.L., and S.R. interpreted the results. J.G.B.D. and R.L. wrote and edited the manuscript.

## Competing interests

S.R. is the founder and an unpaid consultant to Angiocrine Bioscience. All other authors declare no competing interests.
