## [Peer Review File · Communications Biology]

Reviewers' comments:

Reviewer #1 (Remarks to the Author):

In the manuscript by Duran et al., the authors describe the novel role of Jak3 in regulating bone marrow endothelial cell differentiation and function with specific emphasis on the pro-hematopoietic function of bone marrow endothelial cells as relates to Jak3 expression. The analyses of the molecular and immunophenotypic effects of Jak 3 deficiency on BM ECs were very well performed and the conclusions in this regard are well supported by the data presented. The findings regarding the effects of Jak3 overexpression in BM ECs and loss of Jak3 in the BM niche and the impact of both on HSC function are also compelling and additive to the hematology field. I have some specific comments regarding the manuscript findings and conclusions:

Introduction

1. Authors state that endothelial cell support of ex vivo expansion of hematopoietic stem cells is contact dependent. This is context dependent. Other groups have reported that human HSCs could be expanded in non contact dependent EC cultures, suggesting paracrine activities from ECs (Blood 2002;100:4433-9; Blood 2005;105:576-83).

2. The authors also state that BM EC expansion of hematopoietic stem cells is "organotypic". This is not accurate since other labs have shown that brain derived ECs support human HSC expansion (Nature Medicine 2010;16:475-82).

Figure 1

1D. Jak1 appears to be significantly increased in BM ECs and liver ECs compared to all other tissue ECs. Is it possible that Jak1 is as important as Jak3 to the hematopoietic supportive activity of ECs and have the authors tested whether loss of Jak1 in liver ECs has any effect on liver EC capacity to supportive hematopoietic cell expansion in vitro?

Figure 2F. The images shown convincingly indicate that the VECadherin+ vessels shown are largely Sca-1-negative, whereas in Jak3^{-/-} mice, the VECadherin+ vessels are largely Sca-1-bright. Based on several papers including Xu et al. Nat Comm 2018, this would suggest that absence of Jak3 promotes an arterial endothelial cell predominance in the BM (at least as represented in this Figure panel). The associated discussion in the Results section, referencing Figure S2B flow cytometric analysis, is confusing as written. It is unclear what the significance is of the loss of the CD31+VEcad-dim population in the Jak3^{-/-} mice since the expression of Sca1 appears qualitatively similar in both sets of mice.

Figure 3.

3C. Percentage numbers should be added to the gates shown here for LSK cells. Did the percentage of LSK cells increase or maintain over course of co culture? The percentages of LSK cells after culture should be shown for both groups.

Did the authors evaluate non-contact culture conditions to determine if contact with ECs is necessary in context of Jak3 overexpression?

3D. The colony forming assay results are interesting and suggest that Jak3 overexpression in the ECs may suppresses differentiation of HSPCs in culture compared to control cultures. It might be expected to see increased CFU GEMMs relative to the lineage specific CFCs.

3E. Please add numbers to the representative gates for the percentages shown in the flow plots. The data clearly show that Jak3 overexpression amplifies the sustainment of HSCs with repopulating

capacity compared to the HSCs remaining in the vector control cultures. However, it is not possible to conclude that expansion of HSCs has occurred in the Jak3 cultures unless the limiting dilution analysis includes a comparison with the identical starting dose of cells placed in culture (non cultured starting point HSCs). The comparison of the JAK3 EC cultured progeny with the non expanded identical HSC starting cell dose population would be necessary for any conclusion about HSC expansion. If such comparisons have not been conducted, then the conclusions regarding "HSC expansion" should be modified.

Figure 4.

4A, B. The effect of Jak3 deficiency on lung EC co culture support of LSK cells and CFCs is interesting. Do lung ECs support the expansion of HSCs capable of competitive repopulation in recipient mice and at a level relevant to BMECs?

4C,D. The results of Jak3 deficiency on LSK cell numbers in culture and CFU results are interesting and the authors invoke a possible effect on HSPC proliferation as a way to explain the data. The authors should perform cell cycle analysis or at least proliferation analysis in the EC co cultures with Jak3 deficiency and Jak3 overexpression to provide more clear cellular mechanism explanation for these results.

4F. The data shown from this secondary transplant experiment clearly support the conclusion that Jak3 production in the BM niche is necessary for maintenance of an HSC population capable of competitive repopulation. The experiment does not inform however as to whether the BM ECs are the source of the Jak3 in the niche that is important to HSC maintenance. Did the authors do a co culture study as described in 4C using BMECs to determine if Jak3 deficiency in BM ECs causes a loss of competitive repopulating cells compared to input or vector control cultured HSCs?

Reviewer #2 (Remarks to the Author):

The work by José Gabriel et al. investigates the expression of Janus Kinase 3 (JAK3) and its roles on regulating the LT-HSCs maintenance and expansion. The authors provide several lines of evidence to demonstrated that JAK3 expresses in endothelial cells, especially in bone marrow endothelium (BMEC). They found Jak3 deficiency mice (Jak3^{-/-}) show sinusoidal disruption and vascular phenotypes. Moreover, they find that Jak3 is required for LT-HSCs maintenance and expansion. Overall, the findings in this manuscript are largely novel and interesting; The data are mostly convincing. I have a few questions:

1. This is the first report about JAK3 expression on BMEC. In addition to RNA levels, the validation of JAK3 protein level in BMEC is needed.
2. Jak3^{-/-} show increased Sca-1 signal only in BMEC (Fig 2F), but no other endothelium (Fig S2C and S2D). Importantly, the increased Sca-1 signal (PE) (Lower panel of Fig 2F) is completely overlapped with VE-Cad (PA-647). It seems like the PA-637 signals leaked to Sca-1 channel. The images need to be re-confirmed with Alexa Fluor® 488 and VE-Cad (PA-647).
3. In this study, the authors used Jak3 global knockout mice to study the function of Jak3 in the endothelium. It cannot exclude the contribution of Jak3 from other cell types for observations in this study. Therefore, the function of endothelium Jak3 protein needs to be investigated at least on cell culture models, such as testing the LKS cells expansion on Jak3 knockout HUVEC cells (related to fig 3).

4. The previous study from this group found that endothelial cells support hematopoietic stem cells in Notch-dependent fashion. Does Jak3 deficiency affect Notch signaling activation?

Reviewer #3 (Remarks to the Author):

The manuscript entitled "Endothelial Jak3 expression enhances pro-hematopoietic angiocrine function" by Duran et al. explored the role of Jak3 in endothelial cells (ECs), specifically ECs reside in the bone marrow (BM), in supporting the hematopoietic function and maintenance of hematopoietic stem cells (HSCs). The authors first characterized the expression patterns of Jak3 mRNA in different tissue ECs and different cell types in the BM and found Jak3 to be high expressed in EC-sinusoids. The authors then investigated functional requirement of Jak3 using the previously reported global knock-in of enzymatically inactive Jak3^{-/-} mouse model. Transcriptomic profiling of sorted primary BMECs demonstrated a loss of a number of genes associated with sinusoidal lineage. Phenotypic analysis using flow cytometry and immunofluorescent staining demonstrated altered sinusoidal phenotype i.e. VE-CadhiSca+. Given the changes in BMECs in Jak3^{-/-}, the authors examined the effects of Jak3 in ECs on normal HSC function in vitro and in vivo. Overexpression of JAK3 in human HUVEC system resulted in increased phenotypic LSKs and increased frequency of cells with engraftment capability in a limiting dilution competitive transplant assay. On the other hand, LSKs cocultured with Jak3^{-/-} BMECs resulted in a decrease in frequency of cells with engraftment capability in a secondary transplant limiting dilution assay (LDA). Overall, the study addresses an interesting question and reported a strong in vivo phenotype with Jak3 overexpression and Jak3^{-/-} ECs. The study provided evidences supporting the important contribution of BM niche in maintenance of HSCs. However, the enthusiasm for the manuscript is reduced due to the lack of appropriate statistical analysis and unbiased pathway analysis of RNA-sequencing data. Genes in pathways of interest were picked without any global pathway analysis. No detail on number of biological repeats was provided. There are also not enough experimental details provided in the current version of the manuscript, especially in the in vivo transplantation LDA assay, which is required for a robust interpretation of data.

Below are my comments.

Major points

1. For bulk RNA-seq in Figure 1A and 2A, how many samples (n=?) were used for each condition? Figure 1A and 2B showed FPKM values for several hand-picked genes. Are these average FPKM values? What is FDR/p value of test for differentially expressed (DE) genes showed in Figure 1B? Is there a cut off for "overrepresented gene transcripts" included in Figure 1B?
2. Figure 1C showed a String protein-protein interaction and a vocalno plot showing genes in DE analysis highlighting JAK-STAT signaling pathway. Beside JAK-STAT pathway, are there additional significantly enriched pathways resulting from the KEGG analysis done by the authors? Besides KEGG, have the authors tried additional pathway analysis such as GSEA, GO? Instead of hand pick out JAK-STAT, it would be helpful to learn about the global features of BMECs in an unbiased manner.
3. Similar to the analysis in Figure 1, a complete DE analysis for RNA-seq of Jak3^{-/-} vs. control BMECs should be presented. Instead of picking out a few genes of interest, it is important to characterize the most significant changes when Jak3^{-/-} activity is inactivated.
4. In figure 3E and 4F, the authors showed LDA data using ELDA toolkit. How many conditions i.e. cell

dosages were used in the LDA experiments? In ELDA analysis, a response group must be specified. What was the metric used at the "endpoint flow cytometric analyses" as the "response"? A table summarizing all cell dosages, endpoint data (e.g. engraftment %, how many mice engrafted etc.) and number of mice for each group will help clarify this point.

5. In figure 3 in vivo experiments, engraftment of LSK cells was evaluated in primary transplant setting. In figure 4 experiments, engraftment was evaluated in secondary transplant. What is the engraftment phenotype in the primary transplant? Was there any difference in engraftment or any change in frequency of phenotypic LSKs or HSCs? It is important to provide a thorough examination here as these cells were used as donor cells for the secondary transplant.

Minor points:

1. In Figure 1A and 2B, there are several genes showed with FPKM of less than 1 e.g. 1A: Spn 0.81-0.2; 2B: Frzb 0.09 -0.02. Is this very low level of transcript expression considered robust enough?

2. Several sentences are overclaimed.

There are a few examples:

- "Further, joint overrepresentation of Jak3 and of its obligate signaling partner, Il2rg, suggested downstream activation of this pathway in the bone marrow vasculature". The increase in one gene in RNA-seq is not enough to support the statement that there is an activation of the signaling pathway.

- "BMECs also displayed increased expression of Cxcl12 and Jag1 both contributors to the hematopoietic stem cell (HSC) microenvironment" \diamond no data showing Cxcl12 and Jag1 expression was coupled to this statement.

- "While every member of the Jak family of kinases was present in some quantity in all the vascular beds assayed by bulk RNA-seq (Figs. 1D and S1D), Jak3 appeared as a major contributor to the transcriptomic identity of the bone marrow vasculature in particular". There is no evidence that Jak3 is the driving factor shaping the transcriptomic landscape of BM vasculature since there is no global analysis to compare with other pathways. The authors have to provide such analysis to substantiate the statement or it has to be scaled back or modified.

- "Loss of Jak3 disrupted the phenotype of bone marrow sinusoids at the transcript and protein levels." There is no characterization of the "phenotype" at the protein levels. A flow and IF for 2-3 protein markers is not sufficient to make a statement of the whole "protein levels".

- "Having shown that loss of Jak3 function lead to a dysregulated vascular program in the murine bone marrow, including atypical arterial-venous fate commitment ...". The authors have not provided any evidence for "fate commitment". There is no assay measuring "fate commitment" of Jak3-/- BMECs. Changes in expression of few genes associated to certain lineages are not enough to support such a big statement in cell fate regulation.

3. What was the endpoint (how many weeks post transplantation?) of experiments shown in Figure 3E and 4F?

4. "Interestingly, we only detected bone marrow engraftment in one of 35 secondary transplant recipients from the group subjected to Jak3-/- education. Flow cytometric analyses of multilineage engraftment were run across highest-dose control transplant recipients and the single grafted Jak3-/- niche-educated specimen, revealing no lineage distribution abnormalities between the groups (Fig. S3E).".

\diamond It is confusing as the text mentioned "one of 35 secondary transplant recipients from the group subjected to Jak3-/- education" but in S3E, Jak3-/- show 4 or 5 data dots.

Point-by-point rebuttal – Manuscript #COMMSBIO-20-1602A
“Endothelial *Jak3* expression enhances pro-hematopoietic angiocrine function”

Answers to Reviewer #1 (expertise: niche of hematopoietic stem cells)

1.1. *Authors state that endothelial cell support of ex vivo expansion of hematopoietic stem cells is contact dependent. This is context dependent. Other groups have reported that human HSCs could be expanded in non contact dependent EC cultures, suggesting para crine activities from ECs (Blood 2002;100:4433-9; Blood 2005;105:576-83).*

We thank the reviewer for pointing this out. We have amended the manuscript to indicate that ECs can expand and maintain HSC by deploying angiocrine factors that are either secreted or membrane-bound (see lines 67–69).

1.2. *The authors also state that BM EC expansion of hematopoietic stem cells is “organotypic”. This is not accurate since other labs have shown that brain derived ECs support human HSC expansion (Nature Medicine 2010;16:475-82).*

We agree with the reviewer: all vascular beds are endowed with the ability to expand and maintain HSC *in vitro*. Of note, some vascular beds seem to be more specialized and efficient in this regard (*Stem Cell Reports* 2015;5:881-894). We clarified this point in the current version of the manuscript and included references to the papers mentioned by the reviewer in (1.1) and (1.2).

1.3. [Figure 1D] *Jak1* appears to be significantly increased in BM ECs and liver ECs compared to all other tissue ECs. Is it possible that *Jak1* is as important as *Jak3* to the hematopoietic supportive activity of ECs and have the authors tested whether loss of *Jak1* in liver ECs has any effect on liver EC capacity to supportive hematopoietic cell expansion *in vitro*?

While we found that *Jak1* is more highly expressed in both liver and bone marrow ECs, *Jak3* was only overrepresented in the bone marrow. In addition, *Jak1*^{-/-} mice die within days of birth with no expansion defect reported. To elucidate a vascular phenotype, a floxed mouse would be necessary, which could be the focus of a future investigation.

1.4. [Figure 2F] *The images shown convincingly indicate that the VEcadherin+ vessels shown are largely Sca-1-negative, whereas in Jak3-/- mice, the VEcadherin+ vessels are largely Sca-1-bright. Based on several papers including Xu et al. Nat Comm 2018, this would suggest that absence of Jak3 promotes an arterial endothelial cell predominance in the BM (at least as represented in this Figure panel). The associated discussion in the Results section, referencing Figure S2B flow cytometric analysis, is confusing as written. It is unclear what the significance is of the loss of the CD31+VEcad-dim population in the Jak3-/- mice since the expression of Sca1 appears qualitatively similar in both sets of mice.*

We understand the source of confusion: the references to Fig. S2B (now Supplementary Fig. 5) were incorrect in the submitted manuscript. We have fixed this error and made the language clearer to make sure our point comes across: the “disappearance” of the arteriolar sub-population by flow cytometry appears to be the result of increase *Sca1* expression by the sinusoids (lines 262–264).

1.5. [Figure 3C] Percentage numbers should be added to the gates shown here for LSK cells. Did the percentage of LSK cells increase or maintain over course of co culture? The percentages of LSK cells after culture should be shown for both groups.

We have added percentages to this an all flow cytometry plots in the revised manuscript:

1.6. Did the authors evaluate non-contact culture conditions to determine if contact with ECs is necessary in context of Jak3 overexpression?

Other investigators have previously shown that bone marrow ECs expand HSCs *in vitro* in a contact-dependent manner, through Notch receptor activation (*Cell Stem Cell* 2010; 6:251–264). In the revised manuscript, we acknowledge this and other previous studies that have tackled the question of contact-dependent angiocrine signaling (see answers to reviewer’s comments 1.1 and 1.2), but we believe it falls outside the scope of the present investigation.

1.7. [Figure 3D] The colony forming assay results are interesting and suggest that Jak3 overexpression in the ECs may suppresses differentiation of HSPCs in culture compared to control cultures. It might be expected to see increased CFU GEMMs relative to the lineage specific CFCs.

We appreciate the reviewer’s comment, and though HSPCs co-cultured with JAK3-overexpressing HUVECs do present increased CFU-GEMM potential on average, the difference yielded a *p* value > 0.05 by Student’s *t* test when compared to controls. This may be due to the low *n* (the experiment as designed included only three biological replicates), though our new proliferation and cell cycle experiments (see lines 291–297 and Supplementary Fig. 7b) shows that most hematopoietic cells in experimental and control groups are cycling through the expansion process.

1.8. [Figure 3E] Please add numbers to the representative gates for the percentages shown in the flow plots. The data clearly show that Jak3 overexpression amplifies the sustainment of HSCs with repopulating capacity compared to the HSCs remaining in the vector control cultures. However, it is not possible to conclude that expansion of HSCs has occurred in the Jak3 cultures unless the limiting dilution analysis includes a comparison with the identical starting dose of cells placed in culture (non cultured starting point HSCs). The comparison of the JAK3 EC cultured progeny with the non expanded identical HSC starting cell dose population would be necessary for any conclusion about HSC expansion. If such comparisons have not been conducted, then the conclusions regarding "HSC expansion" should be modified.

We agree with the reviewer and have changed the language from “expansion” to “maintenance” (see line 333). In addition, and as stated in response to comment (1.5), we have added the percentages on the representative flow plots.

1.9. [Figure 4A, B] The effect of Jak3 deficiency on lung EC co culture support of LSK cells and CFCs is interesting. Do lung ECs support the expansion of HSCs capable of competitive repopulation in recipient mice and at a level relevant to BMECs?

We observed that lung ECs support the expansion of HSPCs, as noted in Fig. 4A and B; however, we did not perform a limiting-dilution assay to quantify HSC maintenance following co-culture on lung ECs. Lung ECs express many hallmark angiocrine factors, including Notch receptors and ligands that have been shown to improve hematopoietic expansion in co-culture (*Journal of Clinical Investigation*, 2017;127:4242-4256). In addition, infusion of either bone marrow or lung ECs has been shown to mitigate radiation-induced death, though levels of phenotypic HSPCs were three times lower in mice infused with lung ECs compared to those infused with bone marrow ECs (*Stem Cell Reports* 2015;5:881-894). In sum, we anticipate that HSPC expansion on lung ECs does maintain HSCs, but we are unable to say whether it does to an extent that is comparable to their counterpart’s in the bone marrow.

1.10. [Figure 4C, D] The results of Jak3 deficiency on LSK cell numbers in culture and CFU results are interesting and the authors invoke a possible effect on HSPC proliferation as a way to explain the data. The authors should perform cell cycle analysis or at least proliferation analysis in the EC co cultures with Jak3 deficiency and Jak3 overexpression to provide more clear cellular mechanism explanation for these results.

We have corroborated our initial conclusion by performing expansion experiments paired with proliferation and cell cycle studies to address this comment (see lines 348 through 362 and Supplementary Fig. 8a–d). In short, we observed little to no difference in the proportion of cycling or quiescent cells between WT and *Jak3*^{-/-} expansions on bone marrow or lung ECs. While a sizable portion of the LSK population expanded on BMECs (Supplementary Fig. 8d) was in G1 and most of their counterparts expanded on lung ECs appeared to be actively cycling (Supplementary Fig. 8b), the absolute number of LSK cells in S/G2/M co-cultured on knockout BMECs was about twice as large as controls. Since LSK expanded on knockout BMECs also yielded significantly decreased CFU-GEMM potential (Fig. 4f), we conclude that loss of *Jak3* promotes the proliferation of hematopoietic progenitors in co-culture.

Figure S7

Proliferation and cell cycle analyses for LSK expansions on murine lung and bone marrow ECs.

1.11. [Figure 4F] *The data shown from this secondary transplant experiment clearly support the conclusion that Jak3 production in the BM niche is necessary for maintenance of an HSC population capable of competitive repopulation. The experiment does not inform however as to whether the BM ECs are the source of the Jak3 in the niche that is important to HSC maintenance. Did the authors do a co culture study as described in 4C using BMECs to determine if Jak3 deficiency in BM ECs causes a loss of competitive repopulating cells compared to input or vector control cultured HSCs?*

To address this comment, we have performed an experiment to compare the expansion potential of stromal cells to that of bone marrow ECs. We found that bone marrow ECs have increased HSPC expansion potential, and that bone marrow EC-expanded cells have superior CFU potential than their counterparts expanded on stromal cells. These results appear in Fig. S4 in the revised manuscript.

2.1. This is the first report about JAK3 expression on BMEC. In addition to RNA levels, the validation of JAK3 protein level in BMEC is needed.

We have carried out protein validation in the form of immunohistochemistry experiments to address this comment. Whole femurs from either WT or *Jak3*^{-/-} were probed with Abcam antibody ab203611. As shown below and in Supplementary Fig. 2, Jak3 protein is knocked out in the bone marrow vascular niche in *Jak3*^{-/-} mice. We also observed that Jak3 protein was not present on either WT or *Jak3*^{-/-} bone marrow arterioles, supporting our single-cell transcriptomic analysis and further corroborating that Jak3 is enriched in sinusoidal endothelium, as seen below.

2.2. *Jak3*^{-/-} show increased *Sca-1* signal only in BMEC (Fig 2F), but no other endothelium (Fig S2C and S2D). Importantly, the increased *Sca-1* signal (PE) (Lower panel of Fig 2F) is completely overlapped with VE-Cad (PA-647). It seems like the PA-637 signals leaked to *Sca-1* channel. The images need to be re-confirmed with Alexa Fluor® 488 and VE-Cad (PA-647).

We have repeated this staining using the antibody combination suggested by the reviewer. We observed the same results: WT mice displayed positive Sca1 signal only in bone marrow arterioles, while *Jak3*^{-/-} mice showed Sca1 signal in both bone marrow arterioles and sinusoids. We appreciate this comment greatly, as now the staining results are not due to bleeding of one channel into another. For more details, please see lines 568–569 of the Methods section; Table 1 for updated antibody list; and Fig. 2g, also shown below:

2.3. In this study, the authors used *Jak3* global knockout mice to study the function of *Jak3* in the endothelium. It cannot exclude the contribution of *Jak3* from other cell types for observations in this study. Therefore, the function of endothelium *Jak3* protein needs to be investigated at least on cell culture models, such as testing the LKS cells expansion on *Jak3* knockout HUVEC cells (related to fig 3).

We appreciate the reviewer's question; however, as we show by Western blot, HUVECs in culture don't express *JAK3* at steady state. Please see Fig. 3a, also shown below:

We hope that our HSPC expansion experiment, as explained in (1.11) addressed their concern regarding *Jak3* contribution from other cell types.

2.4. The previous study from this group found that endothelial cells support hematopoietic stem cells in Notch-dependent fashion. Does *Jak3* deficiency affect Notch signaling activation?

Our RNA-seq data from murine bone marrow ECs indicate that Notch ligands *Jag1*, *Jag2*, and *Dll4* are downregulated in *Jak3*^{-/-} compared to WT mice (Fig. 2B). In the reviewed manuscript, we emphasize that altered expression of these transcripts, all of which encode angiocrine factors that contribute to a pro-hematopoietic microenvironment (*Cell Reports*, 2013;4:1022-1034; *Journal of Clinical Investigation*, 2017;127:4242-4256; *Nature*, 2019;569:222-228), suggests that *Jak3* is involved in aspects of HSC regulation. In our manuscript, we go on to show that functional readout. We believe that whether *Jak3* ablation exerts a direct effect on Notch signaling falls outside the scope of our investigation.

3.1. For bulk RNA-seq in Figure 1A and 2A, how many samples (n=?) were used for each condition? Figure 1A and 2B showed FPKM values for several hand-picked genes. Are these average FPKM values? What is FDR/p value of test for differentially expressed (DE) genes showed in Figure 1B? Is there a cut off for “overrepresented gene transcripts” included in Figure 1B?

We appreciate the care with which the reviewer parsed our RNA-seq data and thank them for helping us make the relevant panels clearer and more effective. The numbers of biological replicates (n) for each cell type subjected to RNA-seq are indicated the corresponding figure legend. The FPKM values as shown in the heatmaps on Figs. 1A and 2B were indeed averages of absolute (not normalized) values. While we changed the data presentation in Fig. 2 in the revised manuscript so it no longer includes FPKM number, Fig. 1A (now Fig. 1a to conform with journal style) still does. In that panel, it is our intention to show that the cells we sequenced from the liver, lungs, heart, kidneys, and bone marrow are endothelial, without contamination from other sources. We show that these cells express canonical endothelial surface markers, angiocrine factors, and ETS factors. However, a number of these transcripts are also expressed by hematopoietic cells. For that reason, we show that transcripts that are exclusive to hematopoietic cells are not present by the sequenced cells. We also changed the heatmap to a blue-white-red color scale, which we believe will be more familiar than a black-gray-white color scale. The relevant text in the Results section has also been altered to reflect these changes (see lines 122–123 and 126–127). Finally, the revised manuscript also states the FDR cutoffs (0.05) for the differential expression analyses depicted in Figs. 1b and 2b in the respective figure legends.

3.2. Figure 1C showed a String protein-protein interaction and a vocalno plot showing genes in DE analysis highlighting JAK-STAT signaling pathway. Beside JAK-STAT pathway, are there additional significantly enriched pathways resulting from the KEGG analysis done by the authors? Besides KEGG, have the authors tried additional pathway analysis such as GSEA, GO? Instead of hand pick out JAK-STAT, it would be helpful to learn about the global features of BMECs in an unbiased manner.

We have revamped the Gene Set Expression Analysis (GSEA) analysis in Fig. 1c to include a more comprehensive list of terms retrieved by KEGG pathway analysis (see lines 143–150 and Fig. 1 c). We also included unbiased expression data performed based on this GSEA (see Fig. 1d), showing that the JAK-STAT pathway is not only retrieved by GSEA, but that *Jak3* is among the most highly and uniquely expressed transcripts in and primary bone marrow ECs. Further, the Search Tool for the Retrieval of Interacting Genes/Proteins (STRING) database was used to show that the network of co-expressed genes that contributed to the retrieval of the JAK-STAT KEGG term places *Jak3* as the central node. These computational tools were used in an exploratory manner, to narrow down the field of candidates for investigation. Having found an attractive potential new marker of bone marrow endothelium in *Jak3*, we decided to continue to more functional experiments.

3.3. Similar to the analysis in Figure 1, a complete DE analysis for RNA -seq of *Jak3*^{-/-} vs. control BMECs should be presented. Instead of picking out a few genes of interest, it is important to characterize the most significant changes when *Jak3*^{-/-} activity is inactivated.

We have also revamped the RNA-seq and subsequent GSEA analysis presented in Fig 2. The revised manuscript includes a more in-depth and unbiased differential expression analysis between WT and *Jak3*^{-/-} ECs from the lungs and the bone marrow, respectively (see lines 186 through 202, Fig. 2a–d, and Supplementary Fig. 3a). This includes an unbiased look at global transcriptomic changes upon loss of *Jak3* expression beyond its effect on angiocrine factor regulation.

3.4. In figure 3E and 4F, the authors showed LDA data using ELDA toolkit. How many conditions i.e. cell dosages were used in the LDA experiments? In ELDA analysis, a response group must be specified. What was the metric used at the “endpoint flow cytometric analyses” as the “response”? A table summarizing all cell dosages, endpoint data (e.g. engraftment %, how many mice engrafted etc.) and number of mice for each group will help clarify this point.

We agree with the reviewer and have included within each figure a table with the dosages and number of mice assayed as well as a diagram that highlights the end point of the experiment. The metric used at experimental endpoint (week 20 following limiting-dilution transplantation) was 0.2% CD45.1 chimerism or higher. Please see Figs. 3f and 4g, also shown below:

3.5. In figure 3 in vivo experiments, engraftment of LSK cells was evaluated in primary transplant setting. In figure 4 experiments, engraftment was evaluated in secondary transplant. What is the engraftment phenotype in the primary transplant? Was there any difference in engraftment or any change in frequency of phenotypic LSKs or HSCs? It is important to provide a thorough examination here as these cells were used as donor cells for the secondary transplant.

We observed no immuno-phenotypic difference in chimerism (CD45.1⁺) or lineage differentiation into myeloid (Cd11b⁺Gr1⁺) or lymphoid (CD3⁺ or B220⁺) fates in WT vs. *Jak3*^{-/-} peripheral blood following primary transplantation of WT LKS cells. The bone marrow of these mice also exhibited no difference in proportion of LKS cell content, though the LSK compartment in *Jak3*^{-/-} recipients was mostly host-derived. The revised manuscript makes note

of this observation and clarifies that the number of LKS used in the limiting-dilution secondary transplantation experiment that followed was the same for experimental and control groups. This primary engraftment data is included in Supplementary Fig. 8e and f.

3.6. In Figure 1A and 2B, there are several genes showed with FPKM of less than 1 e.g. 1A: *Spn* 0.81-0.2; 2B: *Frzb* 0.09 -0.02. Is this very low level of transcript expression considered robust enough?

As stated in (3.1), we included absolute FPKM values of *Spn* and other hematopoietic-specific transcripts in Fig. 1A to show that our EC purification method prior to RNA-seq was sound. With regard to the RNA-seq analysis now in Fig. 2d and Supplementary Fig. 3a, the heatmaps therein show values that have been normalized transcript by transcript across groups in order to highlight differential expression patterns. These gene lists were all retrieved by GSEA in an unbiased manner.

3.7. [Overclaimed sentence:] “Further, joint overrepresentation of *Jak3* and of its obligate signaling partner, *Il2rg*, suggested downstream activation of this pathway in the bone marrow vasculature”. The increase in one gene in RNA-seq is not enough to support the statement that there is an activation of the signaling pathway.

We have removed this statement from the revised manuscript.

3.8. [Overclaimed sentence:] “BMECs also displayed increased expression of *Cxcl12* and *Jag1* both contributors to the hematopoietic stem cell (HSC) microenvironment” no data showing *Cxcl12* and *Jag1* expression was coupled to this statement.

We have made a correction and added a reference to Supplementary Fig 3b, left, where the relevant data—showing *Cxcl12* and *Jag2* (not *Jag1*) expression—is shown (see line 433 in the revised manuscript).

3.9. [Overclaimed sentence:] “While every member of the *Jak* family of kinases was present in some quantity in all the vascular beds assayed by bulk RNA -seq (Figs. 1D and S1D), *Jak3* appeared as a major contributor to the transcriptomic identity of the bone marrow vasculature in particular”. There is no evidence that *Jak3* is the driving factor shaping the transcriptomic landscape of BM vasculature since there is no global analysis to compare with other pathways. The authors have to provide such analysis to substantiate the statement or it has to be scaled back or modified.

We have changed this wording in the revised manuscript to state our claim more fairly (see lines 153–156). The text now reads, “While *Jak1*, *Jak2*, and *Tyk2* were present in some quantity in all the vascular beds assayed by bulk RNA-seq, *Jak3* appeared as the main *Jak* family kinase to contribute to the transcriptomic identity of the bone marrow vasculature in particular.”

3.10. [Overclaimed sentence:] “Loss of *Jak3* disrupted the phenotype of bone marrow sinusoids at the transcript and protein levels.” There is no characterization of the “phenotype” at the protein levels. A flow and IF for 2-3 protein markers is not sufficient to make a statement of the whole “protein levels”.

We have removed this statement from the revised manuscript to make our claim more fairly.

3.11. [Overclaimed sentence:] *“Having shown that loss of Jak3 function lead to a dysregulated vascular program in the murine bone marrow, including atypical arterial-venous fate commitment ...”. The authors have not provided any evidence for “fate commitment”. There is no assay measuring “fate commitment” of Jak3^{-/-} BMECs. Changes in expression of few genes associated to certain lineages are not enough to support such a big statement in cell fate regulation.*

We have removed the clause “including atypical arterial-venous fate commitment” from the revised manuscript.

3.12. What was the endpoint (how many weeks post transplantation?) of experiments shown in Figure 3E and 4F?

As stated in (3.4), the revised figures now include this information.

3.13. “Interestingly, we only detected bone marrow engraftment in one of 35 secondary transplant recipients from the group subjected to Jak3^{-/-} education. Flow cytometric analyses of multilineage engraftment were run across highest-dose control transplant recipients and the single grafted Jak3^{-/-} niche-educated specimen, revealing no lineage distribution abnormalities between the groups (Fig. S3E).” It is confusing as the text mentioned “one of 35 secondary transplant recipients from the group subjected to Jak3^{-/-} education” but in S3E, Jak3^{-/-} show 4 or 5 data dots.

We really appreciate the reviewer’s attention to detail. The data in Fig. S3E (now Supplementary Fig. 8h) correspond to the highest-dose secondary-transplanted mice at week 20 following limiting-dilution transplantation. While only one mouse of the 35 subjected to secondary transplantation showed CD45.1 chimerism greater or equal to 0.2%, i.e., above the LDA threshold, other recipients from the highest-dose cohort displayed chimerism under that threshold (see leftmost panel). Supplementary Fig. 8h includes all these mice, as its goal is to show no multi-lineage differentiation defect in our graft. The revised manuscript clarifies this point in the corresponding figure legend. We thank the reviewer for pointing this out and helping us make the text clearer.

REVIEWERS' COMMENTS:

Reviewer #2 (Remarks to the Author):

All concerns are addressed

Reviewer #3 (Remarks to the Author):

The authors had addressed all my concerns in the revision. The revised manuscript is much improved and comprehensive. I do not have any further comment. I recommend publication of the manuscript.